# A fully automated high-throughput workflow for 3D-based chemical screening in human midbrain organoids

Henrik Renner[1], Martha Grabos[1], Katharina J Becker[1,2], Theresa E Kagermeier[1,2], Jie Wu[3,4], Mandy Otto[1,2], Stefan Peischard[5], Dagmar Zeuschner[6], Yaroslav TsyTsyura[7], Paul Disse[5], Jürgen Klingauf[7], Sebastian A Leidel[3,4], Guiscard Seebohm[5], Hans R Schöler[1,2]*, Jan M Bruder[1]*

[1]Department for Cell and Developmental Biology, Max Planck Institute for molecular Biomedicine, Münster, Germany; [2]Westfälische Wilhelms-Universität Münster, Münster, Germany; [3]Max Planck Research Group for RNA Biology, Max Planck Institute for molecular Biomedicine, Münster, Germany; [4]Research Group for RNA Biochemistry, Department of Chemistry and Biochemistry, University of Bern, Bern, Switzerland; [5]Department of Cardiovascular Medicine, Institute for Genetics of Heart Diseases, University Hospital Münster, Münster, Germany; [6]Electron Microscopy Unit, Max Planck Institute for molecular Biomedicine, Münster, Germany; [7]Cellular Biophysics Group, Institute for Medical Physics and Biophysics, Westfälische Wilhelms-Universität Münster, Münster, Germany

*For correspondence:
office@mpi-muenster.mpg.de
(HRS);
jan.bruder@mpi-muenster.mpg.
de (JMB)

Competing interest: See
page 32

**Abstract** Three-dimensional (3D) culture systems have fueled hopes to bring about the next generation of more physiologically relevant high-throughput screens (HTS). However, current protocols yield either complex but highly heterogeneous aggregates ('organoids') or 3D structures with less physiological relevance ('spheroids'). Here, we present a scalable, HTS-compatible workflow for the automated generation, maintenance, and optical analysis of human midbrain organoids in standard 96-well-plates. The resulting organoids possess a highly homogeneous morphology, size, global gene expression, cellular composition, and structure. They present significant features of the human midbrain and display spontaneous aggregate-wide synchronized neural activity. By automating the entire workflow from generation to analysis, we enhance the intra- and inter-batch reproducibility as demonstrated via RNA sequencing and quantitative whole mount high-content imaging. This allows assessing drug effects at the single-cell level within a complex 3D cell environment in a fully automated HTS workflow.

## Introduction

A number of uniquely human diseases, including Parkinson's disease, would greatly benefit from a comprehensive human cellular in vitro model that recapitulates key characteristics of midbrain tissues in a high-throughput-compatible format. Three-dimensional (3D) cell culture in general and the ability to generate organ-like aggregates ('organoids') in particular have found a rapid following over the past few years (*Sato et al., 2009*; *Eiraku et al., 2011*; *Nakano et al., 2012*; *Lancaster et al., 2013*; *Quadrato et al., 2017*; *Paşca et al., 2015*; *Iefremova et al., 2017*; *Takasato et al., 2015*; *Dye et al., 2015*; *Takebe et al., 2013*; *McCracken et al., 2014*; *Paşca, 2018*) due to their potential to mimic cellular niches more closely than 2D cell cultures. These approaches promise to develop next-generation high-throughput screens (HTS) that can provide more relevant predictions of drug efficacy and toxicity (*Fatehullah et al., 2016*; *Ranga et al., 2014*; *Fang and Eglen, 2017*;

**eLife digest** In 1907, the American zoologist Ross Granville Harrison developed the first technique to artificially grow animal cells outside the body in a liquid medium. Cells are still grown in much the same way in modern laboratories: a single layer of cells is placed in a warm incubator with nutrient-rich broth. These cell layers are often used to test new drugs, but they cannot recapitulate the complexity of a real organ made from multiple cell types within a living, breathing human body.

Growing three-dimensional miniature organs or 'organoids' that behave in a similar way to real organs is the next step towards creating better platforms for drug screening, but there are several difficulties inherent to this process. For one thing, it is hard to recreate the multitude of cell types that make up an organ. For another, the cells that do grow often fail to connect and communicate with each other in biologically realistic ways. It is also tough to grow a large number of organoids that all behave in the same way, making it hard to know whether a particular drug works or whether it is just being tested on a 'good' organoid.

Renner et al. have been able to overcome these issues by using robotic technology to create thousands of identical, mid-brain organoids from human cells in the lab. The robots perform a series of precisely controlled tasks – including dispensing the initial cells into wells, feeding organoids as they grow and testing them at different stages of development. These mini-brains, which are the size of the head of a pin, mimic the part of the brain where Parkinson's disease first manifests. They can be used to test new drugs for Parkinson's, and to better understand the biology of the brain. Perhaps more importantly, other types of organoids can be created using the same technique to model diseases that affect other areas of the brain, or other organs altogether. For example, Renner et al. also generated forebrain organoids using an automated approach for both generation and analysis.

This research, which shows that organoids can be grown and tested in a fully automated, reproducible and scalable way, creates a platform to quickly, cheaply and easily test thousands of drugs for Parkinson's and other difficult-to-treat diseases in a human setting. This approach has the potential to reduce research waste by increasing the chances that a drug that works in the lab will also ultimately work in a patient; and reduce animal experiments, as drugs that do not work in human tissues will not proceed to animal testing.

*Dutta et al., 2017*; *Ho et al., 2018*; *Chen et al., 2018*; *Friese et al., 2019*) as they may allow better modeling of pathologies with complex interactions of several cell types in specific cellular niches (*Qian et al., 2016*; *Mariani et al., 2015*; *Ogawa et al., 2015*; *Verissimo et al., 2016*; *Vlachogiannis et al., 2018*; *Czerniecki et al., 2018*).

3D culture in the form of spheroids has long been established, especially in the field of cancer biology (*Sutherland et al., 1971*), and used in various HTS applications (*Kelm et al., 2003*; *Senkowski et al., 2015*; *Wenzel et al., 2014*; *Kenny et al., 2015*). While these model systems are already more complex and potentially more physiologically relevant than 2D culture (*Pickl and Ries, 2009*), they display a much simpler, less organ-like 3D cell organization (*Fang and Eglen, 2017*) and do not mimic functional features of the organ as broadly and as closely as organoid tissues. Moreover, many of the 3D-based screens performed so far depend on whole-aggregate-based readouts such as size, morphology, and cell viability (*Vlachogiannis et al., 2018*; *Ivanov et al., 2014*; *Friedrich et al., 2009*; *Ivanov et al., 2015*; *Hou et al., 2018*; *Kang et al., 2015*), which make it challenging to gain mechanistic insights into cells or sub-populations of cells in the context of their niches.

In contrast, complex organoids have emerged as a promising research tool due to their unique resemblance to human tissues, defined by an organ-like architecture composed of different tissue-specific cell types and the capability to mimic organ functions (*Lancaster and Knoblich, 2014*). One structure of particular interest in the context of disease modeling and drug development is the midbrain due to its role in the highly prevalent Parkinson's disease (affecting 2–3% of the population aged 65 and above *Poewe et al., 2017*) and many other developmental disorders (*Barkovich, 2012*; *Doherty et al., 2013*). While the current state of the art allows the generation of different types of midbrain organoids (*Qian et al., 2016*; *Jo et al., 2016*; *Monzel et al., 2017*), the rigorous

standardized organoid production and quantification methods needed for high-throughput screening have been elusive. The established protocols tend not to focus on scalable, homogeneous organoids with quantitatively predictable morphology, cellular composition, and local cell organization. Obligatory extensive manual handling often including cumbersome matrix embedding steps render them challenging for scale-up (*Tong et al., 2018*). Furthermore, common analysis methods (e.g. sectioning and immunostaining, RNA sequencing) do not scale well for HTS applications.

Here, we present a fully automated workflow in a standard 96-well format that combines generation, maintenance, whole mount immunostaining, tissue clearing, and high-content imaging of automated midbrain organoids (AMOs) (see *Figure 1a*). The resulting AMOs are similar to published midbrain organoids with regard to their expression of midbrain-specific markers and cell populations, yet maintain a reproducible homogeneous phenotype. They mimic relevant organ function in the form of spontaneous, highly synchronized neural activity indicating functional cellular coupling across the entire AMO. Their high homogeneity, reproducibility, culture format, and fast development of approximately one month render them ideal for high-throughput screening applications. Moreover, our combined whole mount immunostaining and clearing workflow abolishes the need for labor-intensive tissue sections and allows for quantitative whole mount high-content analysis of entire organoids with single-cell resolution. Our automation of the entire workflow from seeding to analysis in standard plates allows for easy scale-up and implementation into existing screening facilities.

## Results

### Automation enables high-throughput-compatible production of homogenous midbrain organoids

Screening applications require biological systems that operate within predictable physiological parameters. In order to limit cellular heterogeneity during differentiation, we produced human AMOs starting from small molecule neural precursor cells (smNPCs) (*Reinhardt et al., 2013a*), which in turn originate from pluripotent stem cells (PSCs). The neural-restricted developmental potential of smNPCs still allows the self-organization required for the formation of a 3D architecture (*Monzel et al., 2017*; *Di Lullo and Kriegstein, 2017*) during differentiation toward a midbrain fate. To further reduce batch-to-batch variability, we also omitted matrigel embedding and standardized mechanical stresses by using an automated liquid handling system (ALHS). Starting from seeding the organoids, all following steps including maintenance, fixation, whole mount staining, and clearing are performed in a fully scalable automated fashion using a 96-channel-pipetting head in a robotic ALHS (see *Figure 1a*). The resulting AMOs show little intra- and inter-batch variability in size distribution (see *Figure 1b*, average coefficient of variation (CV) within one batch 3.56%; min 2.2%, max 5.6%), morphology (see *Figure 1c*), and cellular composition and organization (see *Figure 2*), making them ideal for HTS-approaches. Furthermore, our workflow generates one aggregate per well, maintained independently from others, thus minimizing batch effects due to paracrine signaling observed in bioreactor-based strategies (*Quadrato et al., 2017*). If paracrine signaling is desired, our workflow can easily accommodate several aggregates per well. The fully automated workflow operates with very high efficiency, retaining 99.7% (standard deviation 0.7%) of samples for automated seeding, aggregation, and maturation steps over 30 days and 96.5% (standard deviation 3.1%) of samples for fixation, whole mount staining, clearing, and transfer to flat bottom imaging plates over 12 days. Lastly, 6.1% (standard deviation 1.3%) of these samples are rejected during high-content imaging for presence of dust, damage, or fibers (see *Figure 1d* and *Supplementary file 1* for source data).

### Automated midbrain organoids express typical neural and midbrain markers and show structural organization

In order to characterize protein localization in our AMOs (>600 μm diameter) and assess the efficiency of their neural/midbrain differentiation at a cellular resolution and in a HTS-compatible manner, we adapted an extended 3D-staining protocol (*Lee et al., 2016*) and combined it with benzyl alcohol and benzyl benzoate (BABB)-based tissue clearing (*Dent et al., 1989*). BABB-based clearing proved to be both the fastest and most efficient method in a comparison of different clearing

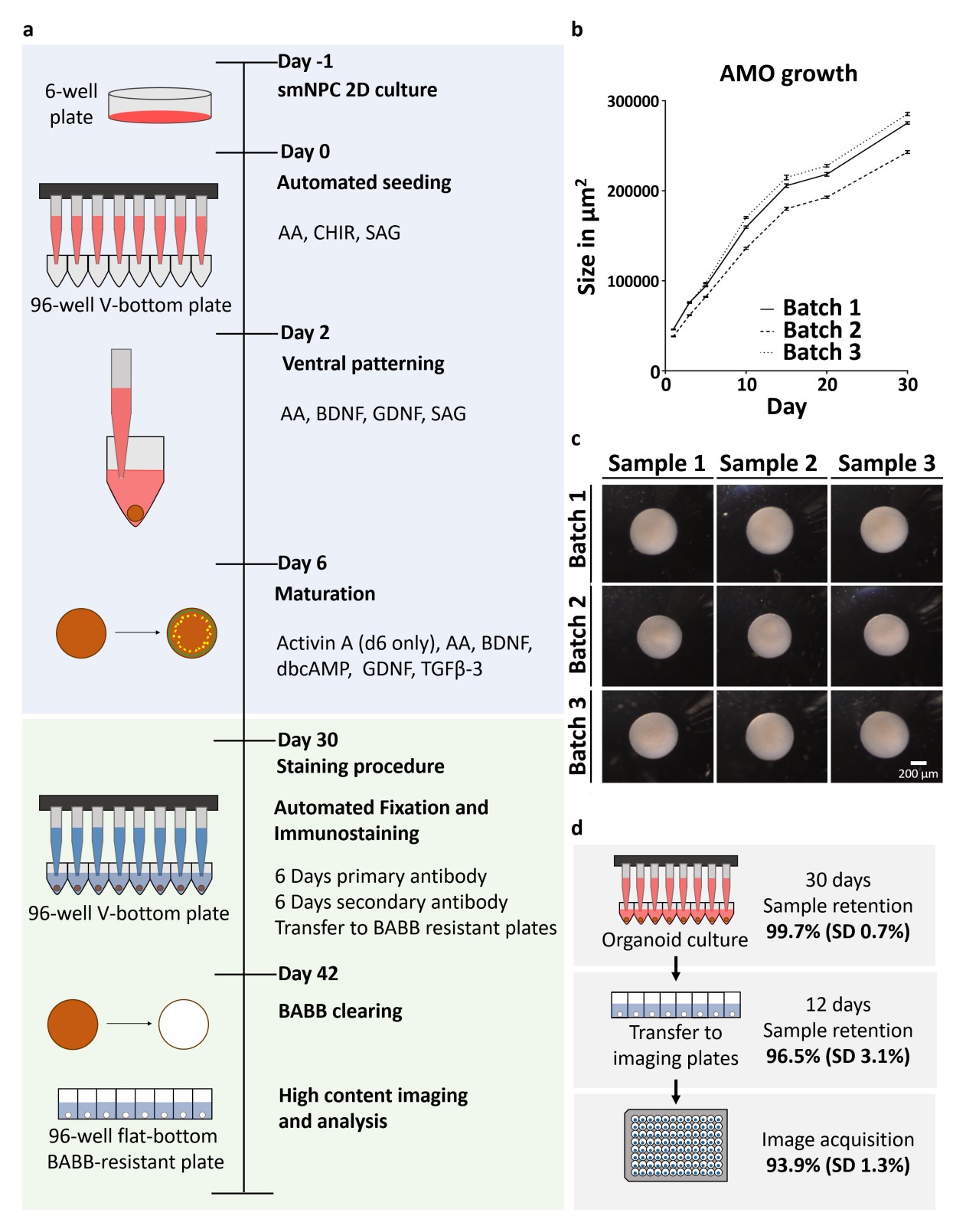

**Figure 1.** Automation enables high-throughput-compatible production and analysis of homogenous midbrain organoids. (**a**) Schematic overview of the automated HTS workflow including organoid generation and optical analysis. (**b**) Measurement of AMO size (area of the largest cross section) reveals low variation and parallel growth kinetics for three batches of AMOs from independently thawed and cultured cells. Error bars represent standard error of the mean (SEM), n ≥ 20 organoids per data point. (**c**) Light microscopy images illustrating the morphological homogeneity of AMOs at day 30 of

*Figure 1 continued on next page*

*Figure 1 continued*

differentiation. Scale bar: 200 μm (**d**) Overview of sample retention at each step of processing. The automated workflow is highly efficient with 99.7% (standard deviation 0.7%) of wells retaining organoids after 30 days of culture. After whole mount staining, 96.5% (standard deviation 3.1%) of samples are successfully transferred to flat bottom imaging plates by the automated liquid handling system. This step can also be repeated without any harm to the samples to further increase efficiency. Finally, 93.9% (standard deviation 1.3%) of samples acquired by the high-content confocal microscope pass image analysis quality control and can be used for downstream analysis. SD = Standard deviation, $n_{Culture}$ = 30, $n_{Transfer}$ = 6, and $n_{Imaging}$ = 3 96-well plates, for a list of the complete source data used to calculate the sample retention efficiency see *Supplementary file 1*. Also see *Figure 1—figure supplement 1*.

The online version of this article includes the following figure supplement(s) for figure 1:

**Figure supplement 1.** Benzyl alcohol and benzyl benzoate (BABB) tissue clearing of organoids is significantly more effective than other protocols.

protocols (see *Figure 1—figure supplement 1*). The combination of whole mount staining and clearing allows the 3D reconstruction of entire AMOs via confocal imaging and enables further detailed 3D quantification and analysis, for example tracing of neurites throughout the whole aggregate, which cannot be performed using typical tissue sectioning procedures (see *Video 1*).

The immunostaining results are depicted as either single confocal optical slices (see *Figure 2a–d, f/g*) or maximum intensity projections (MIP, see *Figure 2e*). Already at day 25, the AMOs contained large numbers of neurons as indicated by the expression of Map2 (*Shafit-Zagardo and Kalcheva, 1998*) (also see *Video 1*), β-tubulin III (TUBB3) (*Leandro-García et al., 2010*) (see *Figure 2e*), and doublecortin (*Gleeson et al., 1999*) (DCX, see *Figure 2c/d*). Presence of high levels of tyrosine hydroxylase (TH, *Figure 2a/b*, also see *Figure 2—figure supplements 1* and *2*), the rate-limiting enzyme in dopamine synthesis (*Nagatsu, 1995*), as well as the expression of the transcription factors Foxa2, Lmx1a, Nurr1, and Pitx3 (*Hegarty et al., 2013*) (see *Figure 2—figure supplements 1* and *2*), are consistent with differentiation toward a dopaminergic midbrain fate. While other neuronal subtypes, specifically GABAergic (vGAT) and glutamatergic (vGLUT1) neurons, are present in AMOs (see *Figure 2—figure supplement 1*), their abundance is low compared to dopaminergic neurons. As commonly seen in all 3D neural cultures, AMOs retain a population of neural precursors identified by the expression of Sox2 (*Ellis et al., 2004*) (see *Video 1*, *Figure 2a/b*, and *Figure 2—figure supplement 2*), Brn2 (*Dominguez et al., 2013*; *Figure 2c/d* and *Figure 2—figure supplement 2*), and the more general neural marker nestin (*Hendrickson et al., 2011*) (see *Figure 2a/b* and *Figure 2—figure supplement 2*).

Over time, AMOs matured further. Expression of synapsin (*Thiel, 1993*) (see *Figure 2—figure supplements 1* and *2*) as well as the presynaptic marker synaptophysin and postsynaptic marker homer (*Tadokoro et al., 1999*) frequently colocalized with each other on Map2-positive neurites (see *Figure 2f*) and indicated the presence of synapses. Since gliogenesis follows neurogenesis in vivo (*Miller and Gauthier, 2007*), we expected the emergence of astrocytes after the initial formation of neurons. Consistently, AMOs contained GFAP and S100b double-positive astrocytes (*Götz et al., 2015*) at later stages (see *Figure 2g*).

In the cortex, neurons form cortical layers with distinct markers. Cortical 3D models recapitulate this layer organization to a degree (*Lancaster et al., 2013*; *Paşca et al., 2015*; *Qian et al., 2016*; *Mariani et al., 2015*; *Bhaduri et al., 2020*). In contrast, the midbrain does not possess the typical layer organization of the cortex, hence published midbrain organoids are devoid of cortex-like layers (*Qian et al., 2016*; *Monzel et al., 2017*). Due to their self-organizing nature, typical published midbrain organoids initially form random local subdomains of organized tissue within the bulk of the organoid, often in the form of rosettes (*Jo et al., 2016*), a hallmark of the very early stages of neural development (*Perrier et al., 2004*; *Elkabetz et al., 2008*). This makes their morphology harder to predict within batches and leads to heterogeneity rendering screening strategies more challenging. We have optimized our AMOs to not form distinct random local subdomains; rather, the different cell types within the AMOs (i.e. neurons, astrocytes, and neural progenitors) self-organize into different concentric zones with distinct cellular orientations spanning the entire organoid (see *Figure 2a–c*). The outermost layer of the AMOs contains few nuclei with a dense, circumferentially oriented layer of TH+/nestin+/DCX+ cell processes. Cellular orientation changes in the underlying zone closer to the core, with TH+ dopaminergic and DCX+ neurons showing a clear radial alignment (see *Figure 2b/c*). The next zone, separating this region of radially organized neurons and the core, contains circumferentially oriented DCX+ neurons and few Brn2+ neural precursors (see *Figure 2b/c*).

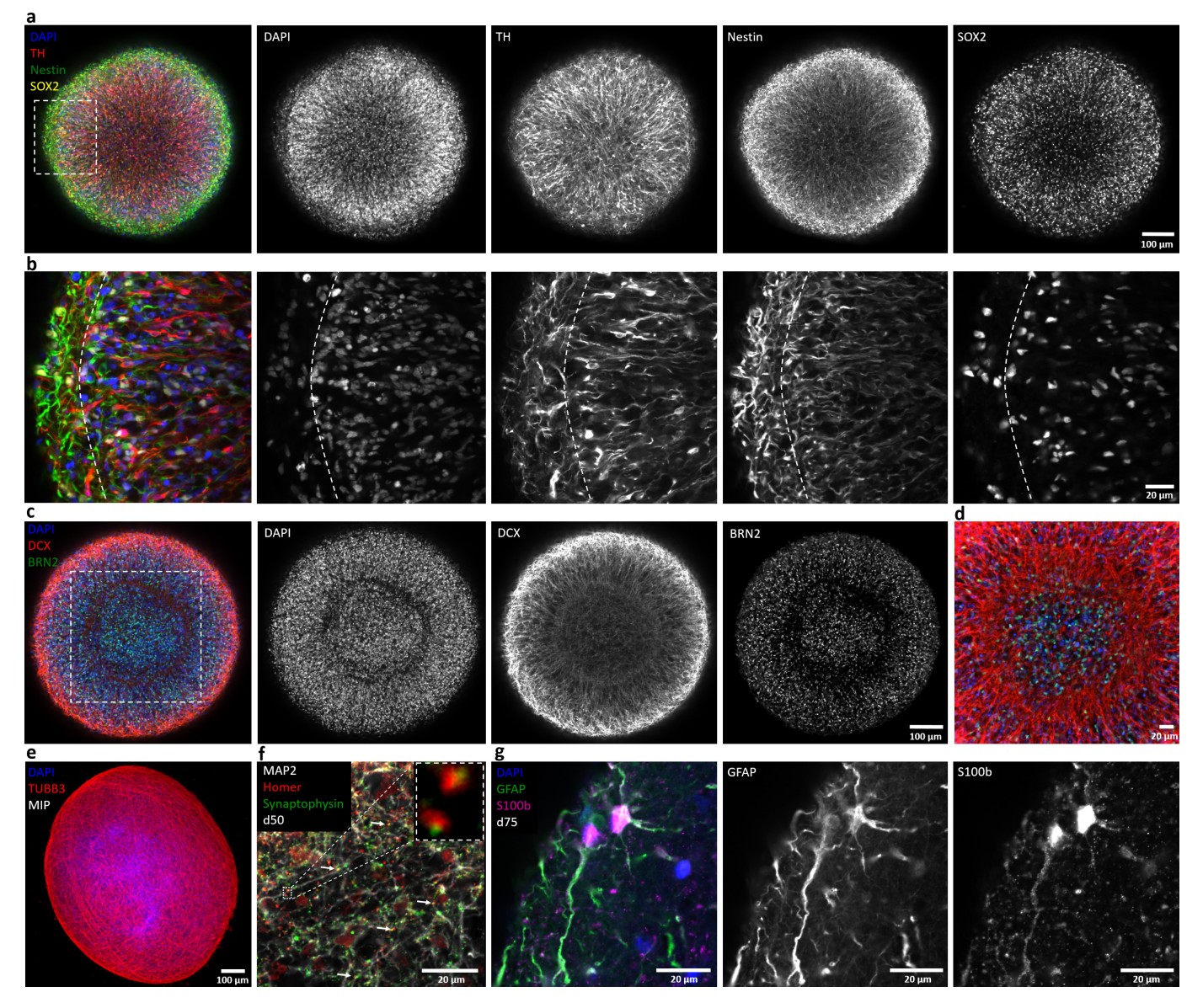

**Figure 2.** Automated midbrain organoids express typical neural and midbrain markers and show signs of structural organization. (**a**) Expression of the dopaminergic midbrain marker TH as well as the precursor markers nestin and Sox2 is evenly distributed throughout the entire aggregate at day 25, as shown by single confocal microscopy slices of tissue-cleared samples. The dotted box indicates the area shown in (**b**). Here, higher magnification of the peripheral aggregate region reveals two different zones with few nuclei but dense, circumferentially oriented neurites distally from the core and radial organization of TH-positive neurons more proximally. (**c**) The expression patterns of DCX and Brn2 further illustrate the organization of neurons (DCX) and neural precursors (Brn2) in the core of AMOs into zones. (**d**) Enlargement (of the dotted box in c) highlighting the circumferential organization of neurons (DCX) surrounding the core. (**e**) Maximum intensity projection (MIP) of fluorescent confocal images showing a dense cellular network expressing the neural marker β-tubulin III (TUBB33) within the AMOs at d25. (**f/g**) Continuing maturation of AMOs is indicated by the presence of synapses marked by the colocalization of the presynaptic synaptophysin and postsynaptic homer on Map2-positive neurites at day 50 (f, top right corner showing enlargement of two synapses without the Map2 channel) and S100b/GFAP double-positive astrocytes at day 75 (**g**). Scale bars: 100 µm (**a, c, e**), 20 µm (**b, d, f, g**). Also see *Figure 2—figure supplements 1–3*.

The online version of this article includes the following figure supplement(s) for figure 2:

**Figure supplement 1.** Automated midbrain organoids express synaptic and midbrain markers.

**Figure supplement 2.** Characterization of AMOs generated from a second, independent patient iPSC-derived smNPC line.

**Figure supplement 3.** Electron microscopy displays ultrastructural morphology of neuronal phenotypes.

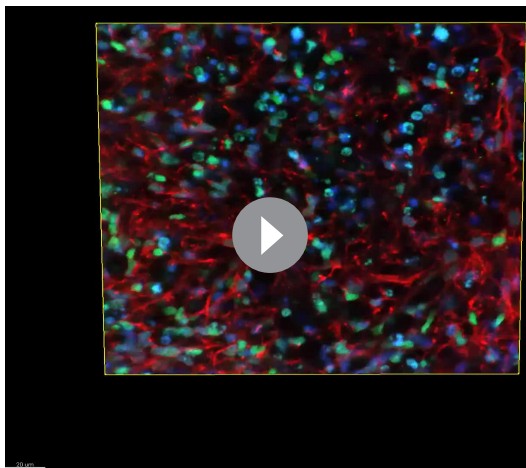

**Video 1.** The combination of whole mount staining and clearing allows confocal imaging of 3D cellular architecture at single-cell resolution. 3D rendering of a confocal stack showing the 3D organization of neural precursors (Sox2, green) and mature neurons (Map2, red) within AMOs. The video shows a cube-shaped volume with edge length of 150 μm. Nuclei were counter-stained with DAPI (blue). AMO at day 25 of differentiation.

https://elifesciences.org/articles/52904#video1

The core itself includes mostly neural precursors and few neurons. Within each zone, the different cell types are homogeneously distributed. In the context of HTS-compatibility, this homogeneity is advantageous, as this trait allows us to extrapolate data even from single medial confocal slices, drastically shortening acquisition times (see *Figure 8—figure supplement 1*). This inter-sample similarity can provide a uniform baseline for compound testing and render self-organized 3D human neural cell niches amenable to HTS strategies.

Ultrastructural analysis of AMOs (see *Figure 2—figure supplement 3*) supported the immunofluorescence data, revealing a dense 3D cell architecture consistent with neuronal cell bodies surrounded by nerve fibers. Analyzing the nerve fibers at a higher magnification revealed regular-spaced neurofilaments and microtubules. Moreover, vesicles with the characteristic size and localization of synaptic vesicles are frequently found within these nerve fibers.

Further quantitative real-time PCR (qPCR) analysis demonstrated increasing expression levels of various neural (DCX, Map2, NEFL, NeuN, TBR2, TUBB3, Syt1), midbrain (TH, NURR1, NKX6-1, EN1, GIRK2, AADC), and glia-specific (GLAST, MBP, S100b) markers at different developmental stages with concomitant decreases in neural precursor markers (Brn2, nestin, Pax6, Sox1, Sox2), confirming neural maturation toward a midbrain identity over time (*Figure 3*). Finally, we replicated these immunostaining and gene expression results with a second independent cell line ('AMO line 2', see *Figure 2—figure supplement 2* and *Figure 3—figure supplement 1*), demonstrating the applicability of our workflow to cells with different genetic backgrounds and origin.

## Calcium imaging and electrophysiological analysis reveal spontaneous and synchronized activity throughout entire organoids

To assess functional coupling of individual cells within the AMOs we first performed Fluo-4 acetoxymethyl ester (AM)-based calcium imaging, which can be used as a readout for spiking activity of neurons (*Grienberger and Konnerth, 2012*). In addition to frequent spontaneous activity of individual cells, we observed aggregate-wide synchronous and periodic calcium spikes (see *Video 2*) in all analyzed AMOs. To characterize this behavior further, we defined different regions of interest (ROIs) and assessed the change in fluorescence intensity over time in each region (see *Figure 4*). Measuring the entire AMO reveals two consecutive spikes in Fluo-4 brightness, with a period of approximately 30 s (see *Figure 4a*). When we subdivided the measured area into four quadrants, we observed synchronized spiking activity in all four resulting ROIs (see *Figure 4b*). This parallel activity pattern could be found at many structural levels of the AMO, even for single cells (see *Figure 4c/e*). Changing the time scale revealed additional levels of synchronicity between selected single cells, in addition to aggregate-wide spikes (see *Figure 4d/e*). Considering the calcium-imaging analyses along with the existence of synaptic vesicles on the ultrastructural level (see *Figure 2—figure supplement 3*), the verification of synapses via immunostaining (see *Figure 2f* and *Figure 2—figure supplements 1* and *2*) as well as synaptotagmin 1 (Syt1) via qPCR (see *Figure 3* and *Figure 3—figure supplement 1*), our results support the presence of functionally coupled and spontaneously active neurons within the AMOs. The synchronous spiking patterns suggest that not only a small number of neurons but, in fact, the entire aggregate may be functionally connected. Large-scale synchronous bursting behavior can also be observed in several developing brain regions in vivo (*Ben-Ari, 2001*) and in brain slices in vitro (*Silva et al., 1991*).

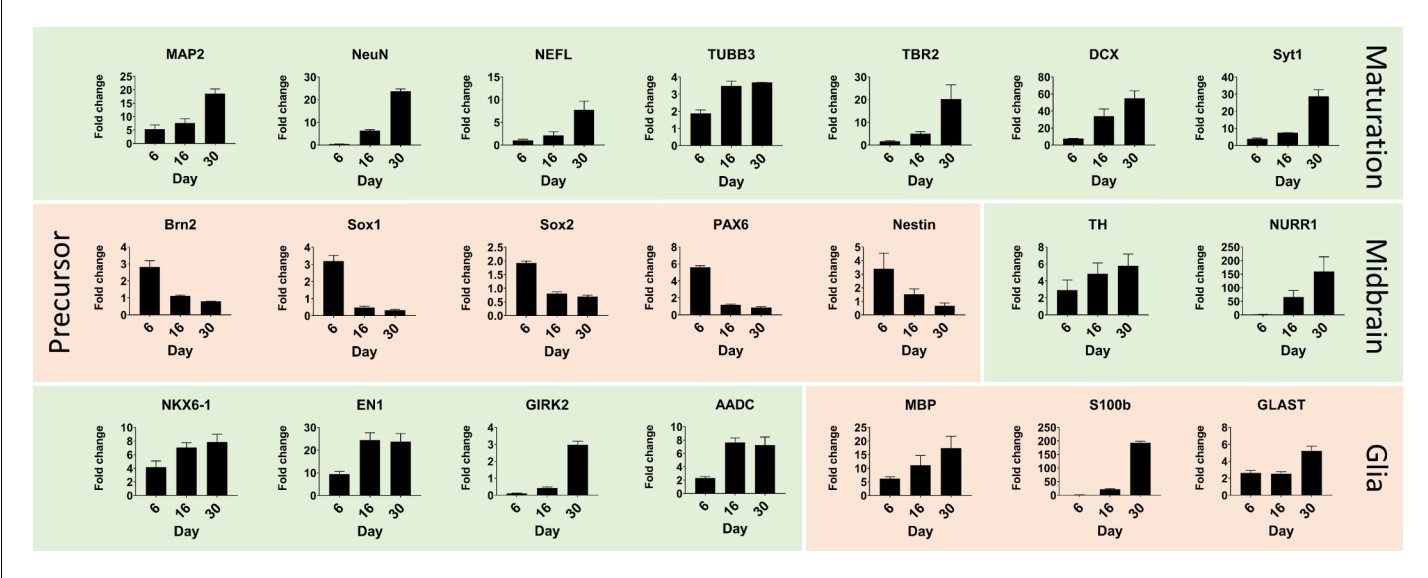

**Figure 3.** Quantitative real-time PCR shows maturation of automated midbrain organoids over time. Changes in gene expression during the development of AMOs shown by qPCR. AMO's continuing maturation is indicated by the increase of neural maturation (MAP2, NeuN, NEFL, TUBB3, TBR2, DCX, Syt1), midbrain (TH, NURR1, NKX6-1, EN1, GIRK2, AADC), and glia (MBP, S100b, GLAST) as well as the decrease of neural precursor (Brn2, Sox1, Sox2, Pax6, nestin) markers over time. (n = 3, error bars = SEM). Also see *Figure 3—figure supplement 1*.

The online version of this article includes the following figure supplement(s) for figure 3:

**Figure supplement 1.** Comparative quantitative real-time PCR analysis between AMOs from two different cell lines and hiPSC organoids confirms correct differentiation toward their respective fates.

Given the high reproducibility of synchronized calcium activity across all tested samples, we decided to evaluate the feasibility of using it as a functional readout in screening settings. Thus, we performed Fluo-4 AM-based calcium imaging on younger (day 35) AMOs and measured the resulting fluorescence signals on a standard plate reader. While they showed a shorter periodicity than the older samples, all tested AMOs displayed distinct peaks in fluorescence intensity resembling the synchronous activity patterns seen during spinning disk microscopy analysis (see *Figure 4f*), also in a second independent AMO line (see *Figure 2—figure supplement 2*). Treatment with the known calcium channel blocker cobalt(II) chloride completely abolished these peaks (see *Figure 4g*). Since the synchronous calcium activity of the AMOs and its modulation by inhibitors can be measured easily via HTS-friendly standard plate readers or specialized FLIPR Ca imagers (*Sirenko et al., 2019*), AMOs may be a promising 3D model of human neural activity that allows directly assessing midbrain related organ functions in HTS.

Multielectrode array (MEA) measurements revealed spontaneous electrical activity in 35-day-old AMOs (see *Figure 4h/i*). The field

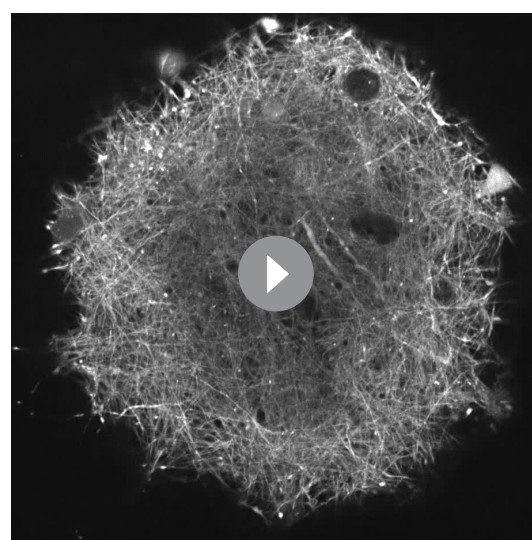

**Video 2.** Automated midbrain organoids display spontaneous and aggregate-wide synchronized calcium activity. Single plane spinning disc confocal time lapse series showing fluctuations in Fluo-4 AM fluorescence of a near-surface tangential optical slice. Images were acquired at 10 Hz for a total of 4 min. Changes are quantified in *Figure 4*. Representative video of n = 5 organoids with similar synchronized activity patterns.
https://elifesciences.org/articles/52904#video2

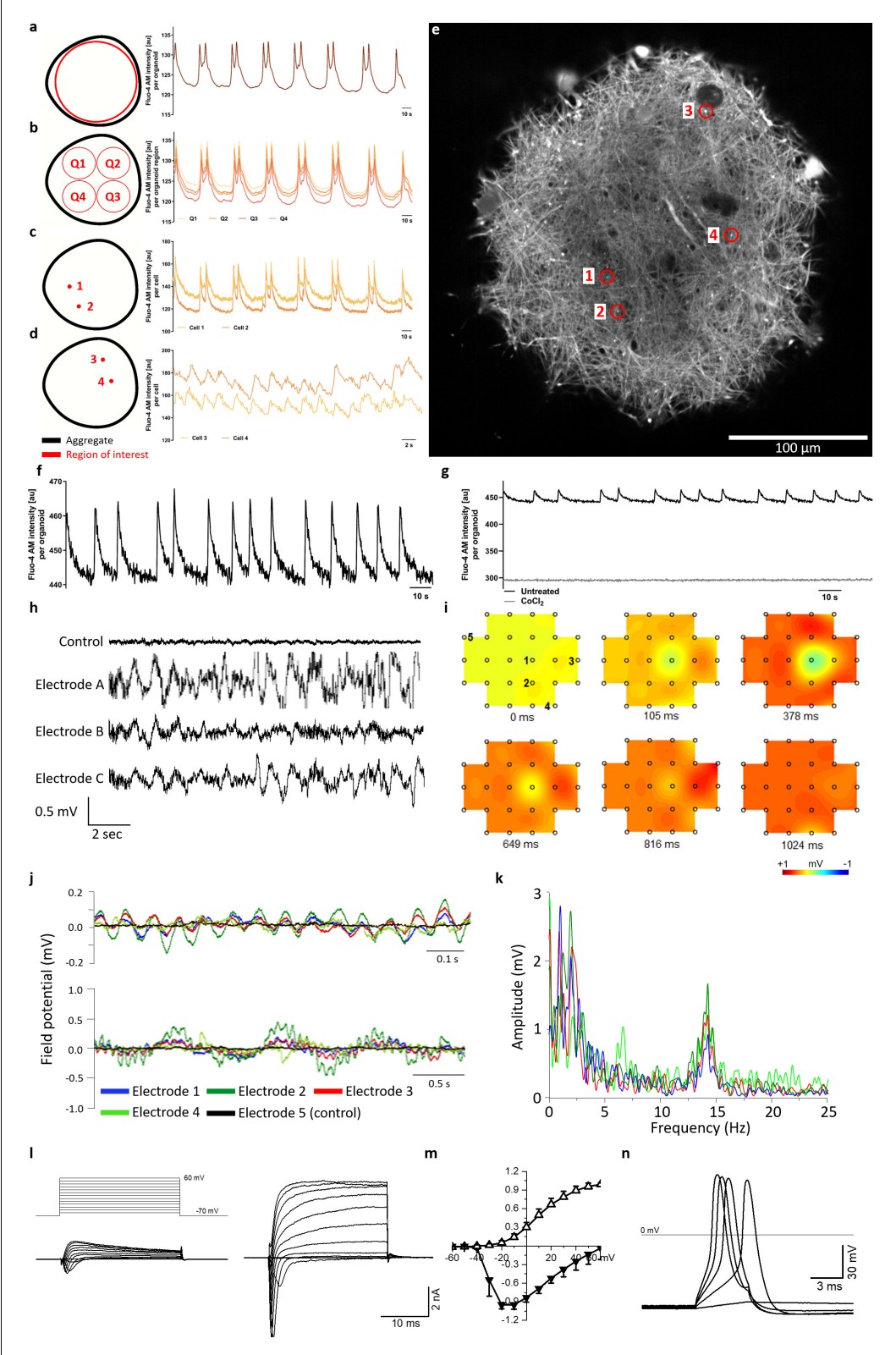

**Figure 4.** Calcium imaging reveals spontaneous and synchronized activity throughout entire organoids. (**a**) AMOs show spontaneous, aggregate-wide spikes of calcium activity. (**b**) Division of the optical cross-section into quadrants shows that this calcium activity is occurring synchronously throughout the entire aggregate. (**c**) This synchronous activity pattern can be found down to the level of single spots. (**d**) Even distant active regions show additional levels of synchronized activity faster than the aggregate-wide spikes. (**e**) Single tangential fluorescent confocal slice indicating the position of

*Figure 4 continued on next page*

*Figure 4 continued*

spots measured in (c and d), also illustrating the dense network of active cells within the AMOs (day 93). For calcium dynamics, please refer to *Video 2*. Scale bar = 100 µm. (f/g) Calcium dynamics with similar activity patterns could also be detected in younger (day 35) AMOs using only a standard plate reader. Treatment with cobalt(II) chloride (CoCl2) completely abolished these spikes. (h) Spontaneous electrical activity of an AMO recorded via MEA at three electrodes (a–c) in the vicinity of the aggregate compared to background activity at a distant electrode (control). All MEA recordings were done with AMOs differentiated for 35 days and n = 3. (i) Field potential map of an AMOshowing the active area and location of electrodes. The distance between electrodes is 300 µm. (j) The field potentials recorded at electrodes 1–4 (j) form synchronous electrical waves at 14 Hz (upper panel) and 1 Hz (lower panel). (k) Fast Fourier Transformation (FFT) based on the data shown in j. (l) Representative recordings of transmembrane currents from two different cells elicited by stepping the membrane potential from −70 to +60 mV in 10 mV increments (schematic of stimulation in the left panel above). The scale bar is common for both recordings, AMOs at day 198. (m) Normalized current-voltage relationship of inward (at peak) and outward (at the end of voltage step) components of transmembrane currents averaged for all neuron-like cells (n = 29). (n) Representative recording of evoked AP's in response to current injections from the cell shown in (l) on the right. Also see *Figure 4—figure supplement 1* and *Figure 2—figure supplement 2*. The online version of this article includes the following figure supplement(s) for figure 4:

**Figure supplement 1.** AMOs display typical neuron-like electrical activity as early as day 25.

potentials of several electrodes in proximity to a 35-day-old AMO (see *Figure 4i*) oscillated in synchrony over time with two concurrent main frequencies at 1 Hz and 14 Hz (see *Figure 4j*, upper and lower panels). *Figure 4k* shows a Fast Fourier Transformation (FFT) based on the data shown in *Figure 4j*. Uncoordinated single-cell activity can hardly account for such robust and spatially long-ranging electric field oscillations. Rather, the concurrent and covariant signals at disparate electrodes support a widespread, synchronized electrical activity encompassing the entire AMO. Taken together, this data further supports the functional coupling of entire AMOs indicated by the calcium imaging experiments.

Finally, we characterized the electrophysiological properties of single cells from the AMOs using voltage patch-clamping. A stepwise increase of the membrane holding potential from −70 to +60 mV with 10 mV increments elicited transmembrane currents that consisted of a fast-activating, fast-inactivating inward current followed by a slower activating, slowly deactivating outward current ranging from a few hundred pA to several nA (see *Figure 4l*). The I–V curves of both currents are typical for sodium inward and potassium outward currents through voltage-gated channels (see *Figure 4m*; *Reinhardt et al., 2013a*; *Simard et al., 1993*; *Cummins et al., 1994*; *Reinhardt et al., 2013b*). Furthermore, the current-clamp recordings demonstrated that these cells generated action potentials (APs) in response to current injections (see *Figure 4n*). The average membrane potential of the recorded cells was −41.9 ± 15.2 mV (n = 29). These typical excitable, neuron-like electrophysiological properties could be detected as early as day 25 (see *Figure 4—figure supplement 1*) and in 29 of the 62 recorded cells. The rest of the cells possessed only outward currents of a few hundred pA by stepping to +60 mV (see *Figure 4—figure supplement 1*) and were unable to generate APs in response to current injections. These may represent other cell types present in AMOs like astrocytes and neural precursors.

## RNA sequencing supports differentiation toward a human midbrain-like fate and homogeneous, predictable gene expression of automated midbrain organoids

To characterize AMOs on the level of global gene expression, we performed RNA sequencing of single organoids from three independent batches of AMOs (i.e. cells were separately thawed, seeded, and cultured) and compared the results with published RNA sequencing data sets of primary human tissues (*Roost et al., 2015*) and established midbrain organoids (*Jo et al., 2016*). Consistent with successful neural differentiation, AMOs were most similar to the brain and spinal cord in a panel with data from 21 human fetal tissues (see *Figure 5a*). Moreover, AMOs also correlated well with published data sets from different midbrain(-like) samples including primary human tissue (see *Figure 5b*). On a global gene expression level, AMOs more closely resembled the primary human midbrain samples than published midbrain organoids (AMOs = correlation 0.78, published midbrain organoids = 0.72, see *Figure 5b*). We also included publicly available RNAseq data from three prenatal human cortex samples (*Jaffe et al., 2015*) in the comparison as non-midbrain controls. As expected, the cortical samples showed high correlation with each other but less with the midbrain samples (see *Figure 5b*; for further comparisons between AMOs and cerebral organoids via the

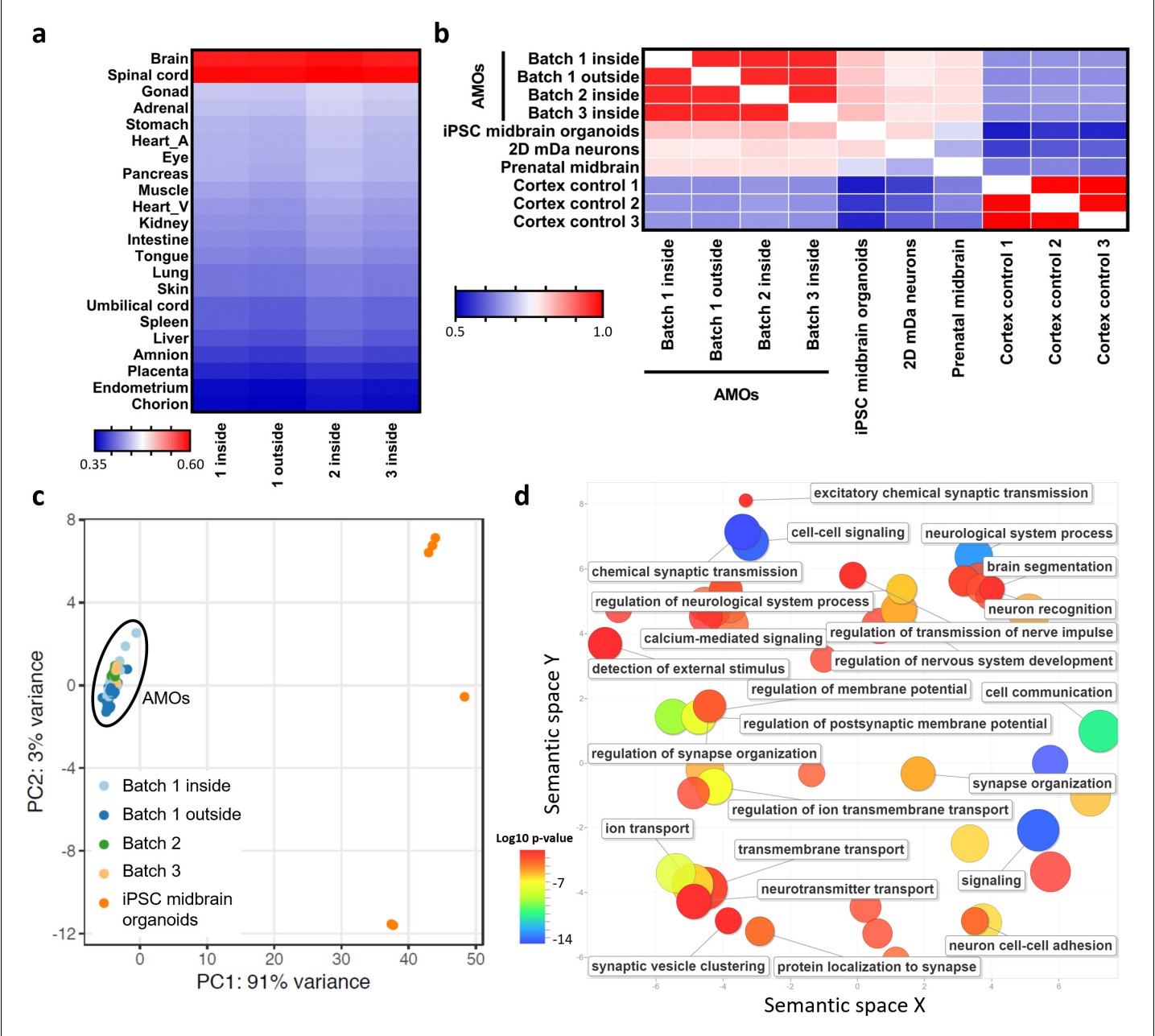

**Figure 5.** RNA sequencing supports differentiation toward a human midbrain-like fate and homogeneous, predictable gene expression of automated midbrain organoids. (a/b) The global gene expression of AMOs correlates with that of fetal human brain and spinal cord tissue (a) as well as published midbrain organoids, 2D dopaminergic (DA) neurons, and prenatal midbrain (b). Shown are heatmaps of the correlation between RNA sequencing data from three independent batches of AMOs and published data sets from either 21 fetal tissues (**Roost et al., 2015**) (a) or of midbrain(-like) origin (**Jo et al., 2016**) (b). (c) AMOs from three independent batches cluster more closely together than published iPSC-derived midbrain organoids (**Jo et al., 2016**) in a PCA plot based on RNA sequencing data. There is no apparent difference between AMOs from the outside or inside of the plate. n = 8, except $n_{1\ outside}$ = 18, $n_{1\ inside}$ = 30, and $n_{iPSC\ midbrain\ organoids}$ = 6. (d). GO term analysis reveals that most genes upregulated in the AMOs (compared to the published midbrain organoids from (b) and (c), with log2 fold change > 2) are related to neuronal and synaptic activity. Visualization via REVIGO (**Supek et al., 2011**), grouping GO terms based on semantic similarity. Each GO term is represented by a circle where the circle size indicates the number of genes included in the term and colors show the significance of enrichment of the term. Also see **Figure 5—figure supplement 1**.

The online version of this article includes the following figure supplement(s) for figure 5:

**Figure supplement 1.** RNA sequencing reveals less intra- and inter-batch variability in AMOs compared to established cerebral organoids.

protocol by *Lancaster et al., 2013*. see *Figure 5—figure supplement 1*). Taken together, AMOs resemble published midbrain organoids as well as primary human midbrain tissue at the level of global gene expression.

Since homogeneity and reproducibility are crucial for screening applications, we next examined the variance of AMOs on the gene expression level and compared it to that of published midbrain organoids (*Jo et al., 2016*). This revealed that AMOs were consistently more reproducible within and between different batches than current midbrain protocols, as illustrated by the principal component analysis (PCA) plotted in *Figure 5c*. The AMOs from three independently thawed and cultured batches (n = 64 separately processed single organoids in total) clustered much more closely together than the published midbrain organoids (n = 6). This further underlines the utility of AMOs as a 3D cellular platform for HTS strategies. In screening settings, the wells at the edges of plates often display different readouts than those located toward the center of the plate ('edge-effects') (*Malo et al., 2006*). Therefore, we sequenced half of a 96-well plate for one AMO batch and tested for differences resulting from well location within the plate (group 'one inside' = center of the plate vs. 'one outside' = edge in *Figure 5a–c*). Importantly, in the PCA plot the AMOs clustered independently of their position on the plate (*Figure 5c*) and the groups also showed no apparent differences in any of the other analyses (*Figure 5a/b*), indicating that AMOs exhibit no measurable edge effects at the global gene expression level and further substantiating the high reproducibility of our protocol.

To further investigate the differences between AMOs and established midbrain organoid protocols, we performed gene ontology (GO) (*Ashburner et al., 2000*; *Supek et al., 2011*) analysis of the genes significantly upregulated ($p_{adj.} < 0.05$) in AMOs compared to the previously used published midbrain organoids (*Jo et al., 2016*). This analysis yielded almost exclusively GO terms connected to neuronal and synaptic activity (*Figure 5d*; for a complete list of GO terms see *Supplementary file 2*). Consistent with the previously described synchronous activity patterns (see *Figure 4*), this further illustrates the physiological relevance and efficient neural differentiation of AMOs.

## Automated whole mount immunostaining is highly quantitative and reveals homogeneity of automated midbrain organoids

While immunofluorescence-based screening-compatible techniques of whole 3D aggregates have been reported, they can only detect cells in the outer layers of large organoids (*Vergara et al., 2017*), or they use small aggregates of approximately 100 µm diameter (*Verissimo et al., 2016*) or cystic organoids (*Czerniecki et al., 2018*), both of which can be penetrated by antibodies and fluorescence illumination more easily. In contrast, our workflow is custom-tailored for automation and allows the quantification of entire dense, large-scale aggregates (>800 µm diameter) with single-cell resolution and high sensitivity, as highlighted by a dose-response assay for 3D cellular detection (see *Figure 6a*). We mixed cells labeled with CellTracker deep red dye with unlabeled cells at known proportions, aggregated them to sizes similar or exceeding that of AMOs (750 µm and 950 µm, see *Figure 6a*), cleared them, and then analyzed them on a confocal high-content imaging system. The resulting relationship between the amount of tracked cells and measured brightness was highly linear ($R^2 > 0.99$), illustrating the quantitative nature of our optical HTS 3D whole mount analysis workflow.

Next, we demonstrated the homogeneity of AMOs at the protein level. A fully automated 96-well based whole mount optical analysis (see *Figure 6b* left) illustrated the ability to detect both abundant filamentous structures (neural marker Map2) and nuclear markers (Sox2) in a HTS-compatible manner (see *Figure 6b* right, single slice from one aggregate). Using nuclear markers like Sox2, our technique allowed quantification at single-cell resolution by identifying, counting, and summing the brightness of Sox2+ nuclei for each imaged confocal plane (see *Figure 6c/d/f/h*). Filamentous, abundant signals like Map2 could be quantified throughout 3D aggregates by summing the overall mean brightness for each confocal plane (see *Figure 6e/g*). The comparison of three 96-well plates from independent batches revealed the uniform cellular composition of AMOs within and between independently thawed and cultured batches (see *Figure 6d–g*) (Average $CV_{Sox2} = 5\%$, $CV_{Map2} = 9\%$).

Positional analysis detected effects of plate position (edge effects) for Map2 levels but not Sox2 levels with about 10% reduced Map2 brightness of samples in the center of the plate (*Figure 6—figure supplement 1*) compared with the wells at the edge. Considered together with the absence of edge effects in the RNA sequencing results, this may indicate that only a specific subset of proteins

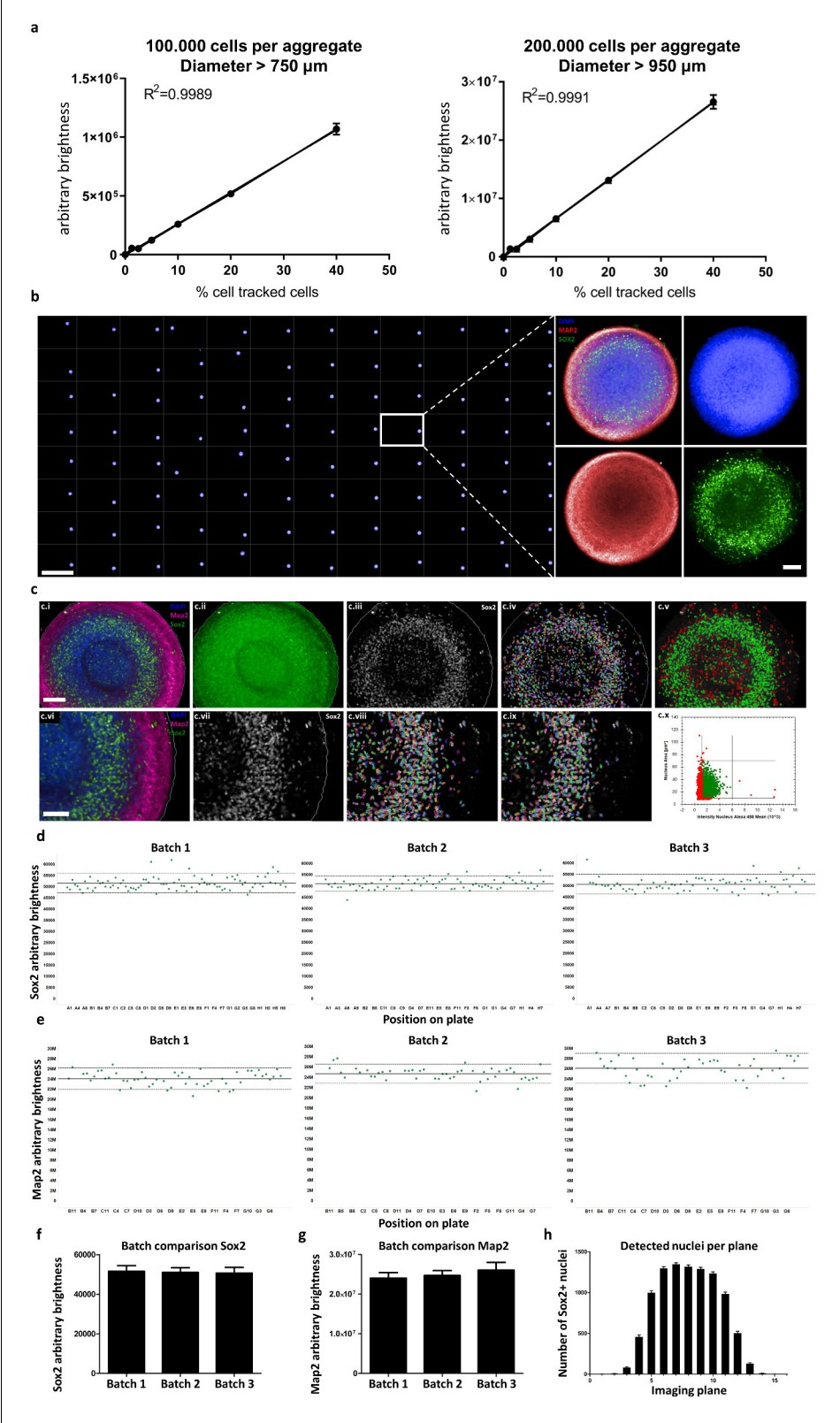

**Figure 6.** Automated whole mount immunostaining is quantitative and reveals high homogeneity of automated midbrain organoids. (a) The optical analysis workflow allows quantification of cell numbers in 3D aggregates. The correlation between the number of fluorescent cells in an aggregate and its brightness measured with our workflow is highly linear ($R^2$ > 0.99) for large-scale 3D aggregates of different sizes (100,000 or 200,000 cells per aggregate, diameter > 750 μm and 950 μm, respectively). n = 3, error bars = SEM. (b) Overview of an entire 96-well plate processed with our HTS-
*Figure 6 continued on next page*

*Figure 6 continued*

compatible optical analysis workflow (left) and an example single plane confocal image of a single AMO illustrating the high cellular resolution achieved with high-content imaging (right). Scale bars: 5 mm left/overview; 100 μm right/enlargement. (c) Visualization of the automated image analysis sequence for the example of Sox2. Images show a single automatically acquired confocal image plane through the center of an AMO. Top row: Overview, with bottom row providing enlarged view. (c.i/vi) Starting image. (c.ii) All three channels summed for aggregate detection. Detected aggregate area overlaid in green. (c.iii/vii) Sox2 channel after sliding parabola treatment to remove background. (c.iv/viii) Sox2 channel with detected nuclei. (c.v) Nuclei selected as Sox2+ according to size and brightness (green) and rejected nuclei (red). (c.ix) Selected nuclei from (h) marked, rejected nuclei unmarked. (c.x) Scatter plot showing nuclear size and brightness distribution and selection thresholds. Scale bars: 100 μm (c.i), top row; 70 μm (c.vi), bottom row. (d–g) AMOs are homogenous with regard to the amount of Sox2 (d/f) and Map2 (e/g) positive cells they contain. In (d and e) each dot represents a single AMO, each graph originating from an independent batch (i.e. cells were separately thawed, cultured and processed). The continuous line represents the mean of all data points on the graph (i.e. Map2/Sox2 content) and the dotted lines correspond to 1.5 confidence intervals. (f and g) Summarize the data of the dot plots as a bar graph. (Error bars = standard deviation, SD). (h) The number of Sox+ nuclei detected in each imaged confocal plane correlates with AMO morphology. The high-content image analysis workflow detects many nuclei where the aggregate diameter is largest (plane 6–10) and fewer nuclei in the first/last planes where it is smaller. (Error bars = SEM). Also see *Figure 6—figure supplement 1*.

The online version of this article includes the following figure supplement(s) for figure 6:

**Figure supplement 1.** High-content imaging analysis reveals edge effects for Map2 but not Sox2.

is altered by edge conditions, while the vast majority of cellular processes is uniform throughout the plate.

## smNPC-derived AMOs are morphologically, structurally, and functionally more homogeneous than automated hiPSC-derived organoids

Since differentiation outcomes and kinetics are known to vary considerably between cell lines, we decided to benchmark our AMOs against a protocol that can be implemented using the same starting cell line as our AMOs and that is adaptable to the same automation and analysis workflow that we established for our midbrain model. In this comparison, smNPC-derived AMOs (line 2) and the hiPSC organoids share the same cell line of origin. We compared our smNPC-derived cultures to hiPSC-derived 3D neural organoids based on a core protocol by *Paşca et al., 2015* (also described in more detail by *Sloan et al., 2018*) with modifications, as they share a number of key traits with our AMOs (for an overview of our automated protocol and the differences to the published original, see *Figure 7—figure supplement 1*). They are self-aggregated and self-organized, and they do not require the addition of an external matrix for proper development. Furthermore, to eliminate any potential bias due to manual handling, we adapted the cortical protocol to our automation pipeline. As a result, any remaining variability did not originate from handling but from stochastic biological processes. For detailed characterization, hiPSC cortical organoids underwent our established automated whole mount staining and clearing procedure (see *Figure 7—figure supplement 2*) as well as qPCR (see *Figure 3—figure supplement 1*), confirming differentiation toward their correct cortical fate in our workflow. Out of twelve full 96-well plates we were, with our workflow, able to generate and maintain cortical organoids in all but one well, where the organoid got lost during the 30 days of automated culture. This further underlines the adaptability and efficiency of our midbrain protocol for other organoid types.

Compared to smNPC-derived AMOs, morphology and zonal arrangement of neural subpopulations in hiPSC organoids varied to a larger degree (see *Figure 7a–d*). Three independently cultured batches of automated hiPSC organoids showed up to a 5-fold higher coefficient of variation in cellular viability and up to a 10-fold higher coefficient of variation for organoid size than AMOs (see *Figure 7g/h*, for individual organoid size and viability data see *Figure 7e/f*).

We performed high-content analyses at the protein level analogous to the data in *Figure 6* and found that the variation of Sox2 and Map2 content of the automated hiPSC-organoids was larger than for AMOs, even when we normalized for strongly variable sizes (see *Figure 7i/j*). The acquisition of high-content data at the same hardware settings for both types of organoids (from the same cell line of origin) also allowed for a direct comparison of cell-type-specific signals. hiPSC-organoids contained a distinctly lower amount of Map2 per area, indicating less efficient/delayed neuronal maturation.

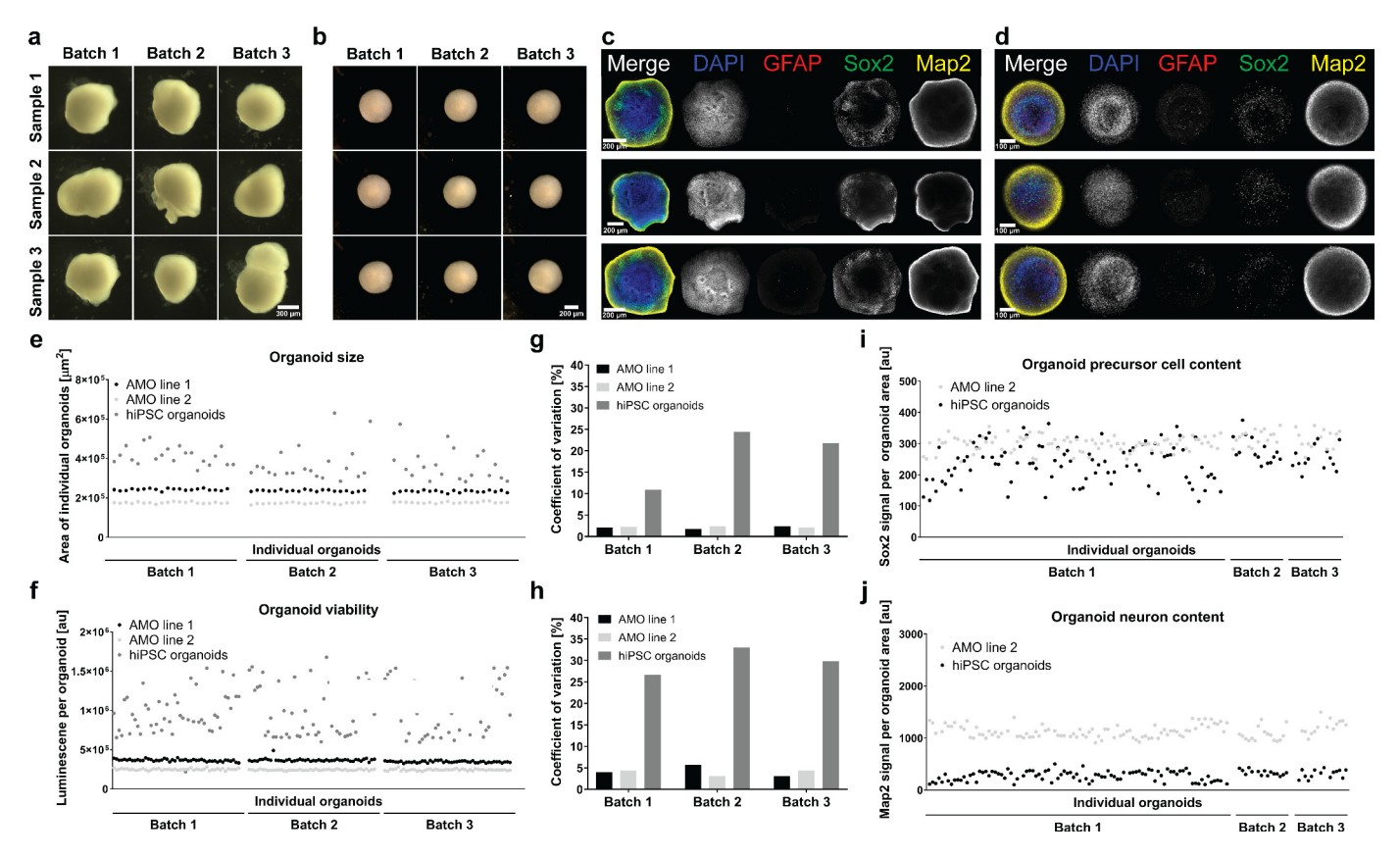

**Figure 7.** smNPC-derived AMOs are morphologically, structurally, and functionally more homogeneous than automated hiPSC-derived organoids. (a/b) Light microscopy images of hiPSC-derived organoids (a) and AMOs (b) generated from the same cell line demonstrating the higher morphological homogeneity of AMOs at day 30 of differentiation.(c/d) Single optical confocal slices of either hiPSC-derived organoids (c) or AMOs (d) at day 30 stained for DAPI, the astrocyte marker GFAP, the neural precursor marker Sox2, and the neuronal marker Map2. The direct comparison illustrates the higher level of structural homogeneity as well as accelerated maturation, especially the earlier emergence of GFAP+ astrocytes in AMOs. Rows depict three samples from one batch. (e/f) Size (area of the largest cross section) and cell viability measurements of individual organoids from three independent batches (per cell line/differentiation protocol) illustrating the high homogeneity of AMOs compared to standard hiPSC organoids. (g/h) Coefficients of variation calculated based on the data shown in (e) and (f). (i/j) Quantitative whole mount staining (see also *Figure 6*) for Sox2 (i), and Map2 (j) showing the higher variability of hiPSC organoids compared to AMOs from the same iPSC line even after normalization to the organoid area. All data gathered from organoids at day 30 of differentiation. Scale bars: 300 µm (a), 200 µm (b/c), 100 µm (d). Also see *Figure 7—figure supplements 1* and *2*.

The online version of this article includes the following figure supplement(s) for figure 7:

**Figure supplement 1.** Overview of the protocol for the automated generation of hiPSC-based organoids and modifications from the published original.

**Figure supplement 2.** The expression of typical neural and cortical markers confirms the correct differentiation of automated hiPSC-derived organoids.

In conclusion, AMOs are more homogeneous with regard to their morphology, cellular structure, size, viability, and protein expression than the automated hiPSC organoids generated from the same cell line and under strictly standardized conditions.

## Automated midbrain organoids possess functional characteristics of midbrain tissue and allow assessment of neural subpopulations for high-throughput screening

A clinically relevant midbrain model requires dopaminergic activity. To further confirm midbrain-specific function of AMOs, we generated electrophysiological data from multielectrode arrays together with specific molecular agonists and antagonists for dopaminergic, GABAergic, and glutamatergic pathways. Functional responses were consistent with a midbrain identity of AMOs (see *Figure 8e/f/*

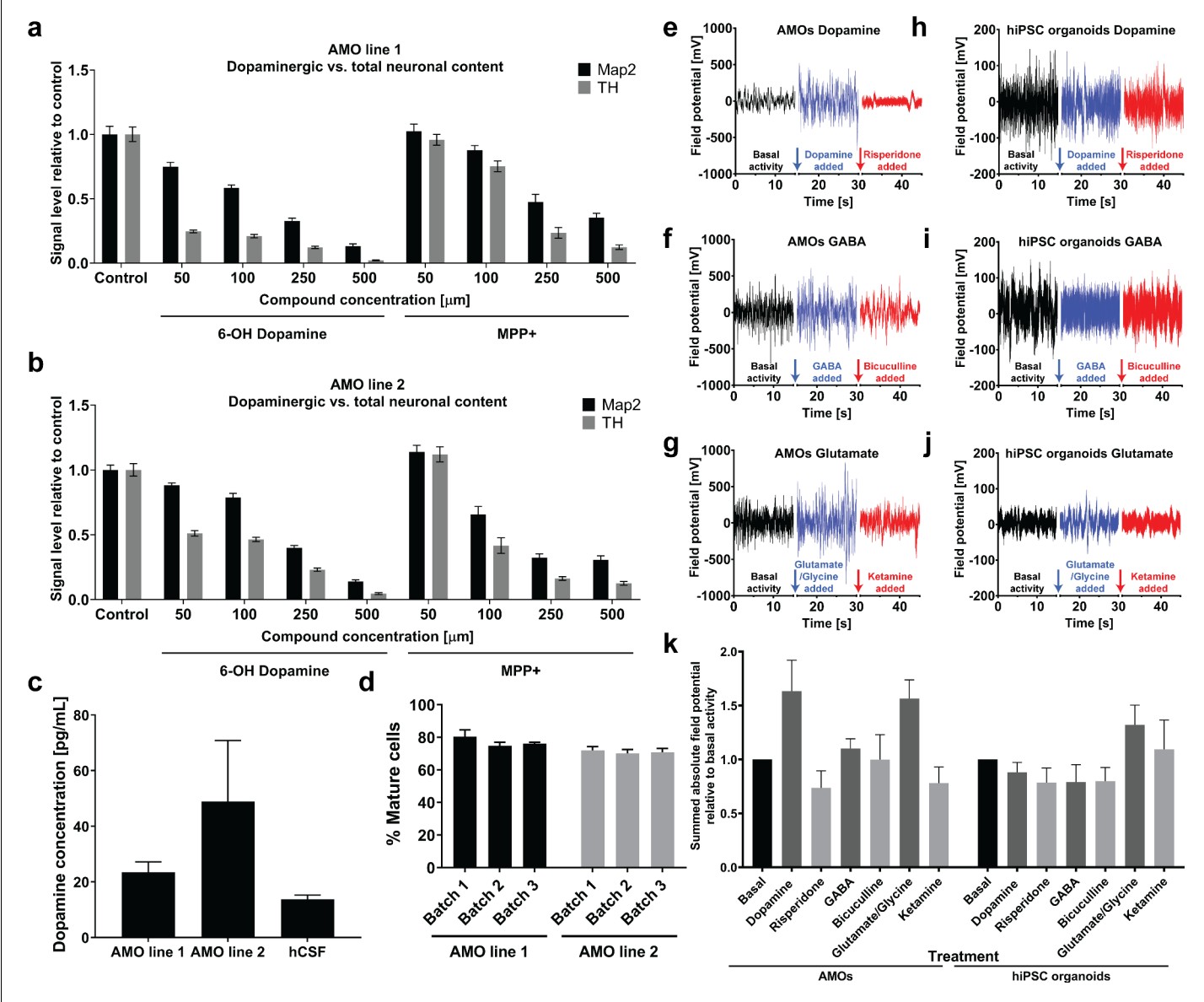

**Figure 8.** Automated midbrain organoids possess functional characteristics of midbrain tissue and allow assessment of neural subpopulations for high-throughput screening. (a/b) The combination of AMOs and our automated whole mount staining and clearing workflow allows the quantification of dopaminergic neuron-specific toxicity in 3D. 6-Hydroxy dopamine and MPP+ specifically ablate TH-positive dopaminergic neurons from the AMOs in a dose-dependent manner and with little variation between replicates and cell lines. n ≥ 6 organoids per data point, Error bars: SEM. Organoids at day 56 of differentiation. (c) 35 days old AMOs secrete dopamine into their cell culture medium under standard culture conditions and without further stimulation, as confirmed by ELISA. The concentration is in the same range as the dopamine levels measured in the cerebrospinal fluid (CSF) of healthy, adult humans as reported by *Goldstein et al., 2012*. $n_{Line\ 1} = 4$, $n_{Line\ 2} = 3$, $n_{hCSF} = 38$, Error bars: SEM. (d) 70–80% percent of cells within the AMOs are negative for the precursor marker Sox2 after 30 days of differentiation. AMO line 1: $n_{Batch1} = 90$, $n_{Batch2} = 16$, $n_{Batch3} = 15$; AMO line 2: $n_{Batch1} = 89$, $n_{Batch2} = 16$, $n_{Batch3} = 14$; Error bars: SD. (e–j) AMOs respond most strongly to dopaminergic modulation and, to a lesser extent, also to glutamatergic modulation while the automated cortical hiPSC organoids are mostly affected by compounds targeting glutamatergic neurons. MEA measurements of individual AMOs (e–g) or cortical hiPSC organoids (h–j) were performed in three stages on the same sample: first, under basal conditions (black line), second, after treatment with an agonist (blue line), and third, after addition of an antagonist (red line). The pharmacological modulators targeted dopaminergic (e/h), GABAergic (f/i), or glutamatergic (g/j) neurons. The gaps in the X-axis represent the addition of the different compounds and the time we allowed for the solution to equilibrate. Shown is the raw signal of one representative example of n = 4. AMOs were at 33 days and hiPSCs at 35 days of differentiation. (k) Quantification of the effects of pharmacological modulation on AMOs and automated hiPSC organoids as measured by MEA. The bar graph shows the sum of the absolute electric field potential oscillations over 15 s of time relative to basal conditions for each modulator. Each bar represents the mean +/- SEM for n = 4 replicates, one representative raw measurement per condition is shown in (e-j). n = 4, except $n_{hiPSC}$ organoids Glutamate = 3; Also see *Figure 8—figure supplement 1*.

*Figure 8 continued on next page*

*Figure 8 continued*

The online version of this article includes the following figure supplement(s) for figure 8:

**Figure supplement 1.** AMOs allow HTS-compatible toxicity evaluation in whole organoids or specific cellular subpopulations.

*g/k*). Electric field oscillations of AMOs responded strongly to the addition of modulators of the dopamine pathway, but yielded more limited responses when targeting the GABA pathway (see *Figure 8e/f/k*). This correlates well with the lower abundance of GABAergic neurons in AMOs as detected by immunostaining (see *Figure 2—figure supplement 1* for a vGAT staining). The electrical activity increases following the stimulation with glutamate agonists and subsequently decreases after addition of the antagonist (*Figure 8g/k*), despite the low abundance of glutamatergic cells (see vGLUT1 staining in *Figure 2—figure supplement 1*). However, this is consistent with previous studies reporting an increase in firing rate and burst-firing for dopaminergic neurons treated with glutamate agonists which can be counteracted by NMDA receptor antagonists (*Meltzer et al., 1997*; *Wang and French, 1993*). Automated hiPSC-derived cortical organoids served as negative controls to validate midbrain-specific MEA data. As expected following cortical differentiation, control hiPSC organoids had a weak response to dopaminergic modulation and reacted most strongly to glutamatergic modulation (see *Figure 8h–j,k*). The overall higher electric field amplitude in AMOs compared to automated hiPSC-organoids further illustrated the accelerated differentiation and faster maturation achieved by our workflow over hiPSC-based organoid protocols of the same age.

To further evaluate cellular maturity as one key factor in physiologically relevant screens, we determined the percentage of Sox2-negative mature cell types within a large number of AMOs via high-content imaging. Testing AMOs from two separate lines with three independently cultured batches each, we detected between 70% and 80% mature cell types after only 30 days of differentiation with very little variation within and between batches and cell lines (see *Figure 8d*).

The maturity and dopaminergic midbrain identity of AMOs was further confirmed by the spontaneous and unstimulated secretion of dopamine into the culture medium at a similar level as found in adult human cerebrospinal fluid (CSF, see *Figure 8c*; *Goldstein et al., 2012*). Sequestered dopamine may constitute an additional readout that is functionally relevant in models of the human midbrain.

To further assess the physiological relevance of our workflow as well as its ability to quantify drug effects in 3D cell cultures, we treated AMOs from two different cell lines with increasing concentrations of toxins specific for dopaminergic neurons, namely 6-Hydroxy Dopamine (6OHD) and 1-Methyl-4-phenylpyridinium (MPP+) (*Reinhardt et al., 2013b*; *Blum et al., 2001*; *Simola et al., 2007*; *Meredith and Rademacher, 2011*). High-content imaging allowed us to discern distinct effects of these nigral toxins on TH-positive neurons compared to general MAP2-positive neurons in AMOs. While both compounds reduced the number of all neurons in a dose-dependent manner, levels of TH-positive dopaminergic neurons decreased more strongly and at lower concentrations (see *Figure 8a/b*), with 6OHD having the more specific effect. Overall, our HTS workflow detected subpopulation-specific effects within dense, whole-mount-stained organoids with very little variation between replicates and between AMOs from separate cell lines.

We next evaluated the use of cleaved caspase 3 (cCasp3) as a general marker for cell death in AMOs for HTS toxicity studies (see *Figure 8—figure supplement 1*). If used judiciously, cCasp3 staining can yield quantitative dose-response curves, for example for the addition of known cell toxicants such as G418 (see *Figure 8—figure supplement 1a, b, d*). When used in co-staining scenarios, this makes cCasp3 potentially attractive to quantify the viability of a wide range of cellular subpopulations, such as Sox2+ precursor cells (see *Figure 8—figure supplement 1c, e*).

However, cCasp3 only labels cells that are currently undergoing apoptosis, and this signal disappears when dead cells are cleared from the tissue. Consequently, we observed that the cCasp3 signal dropped at high concentrations of toxicants (see *Figure 8—figure supplement 1g*), necessitating a careful optimization of assay timing and assay windows. This makes cCasp3 challenging to use for primary screens, where compounds with a wide range of toxicities and kinetics may need to be evaluated in parallel. Other viability assays, such as the ubiquitous CellTiter-Glo 3D assay, provide a much broader dynamic range and are less sensitive to timing, as they quantify living cells instead of a transient signal of dying cells (see *Figure 8—figure supplement 1i*). Unfortunately,

CellTiter-Glo cannot distinguish survival of cellular subpopulations, since it relies on a non-discriminatory lysis protocol. Thus, CellTiter-Glo may be most suitable for primary screens and cCasp3 staining as a follow up to probe effects on specific cellular subpopulations. Another common readout in 3D cell culture screens is aggregate size (*Yao et al., 2020*; *Mittler et al., 2017*; *Kim et al., 2020*). Interestingly, the largest cross-sectional area of organoids did not correlate with G418 doses or CellTiter-Glo survival data in the conditions and time scales tested here, but remained largely constant despite escalating toxin levels (see *Figure 8—figure supplement 1h*). Our automated workflow is compatible with all three types of assay and the results demonstrate that AMOs show lower variance in cell death/survival in toxicity studies than iPSC-organoids derived from the same cell line (see *Figure 8—figure supplement 1g, i*).

Overall, this data highlights the suitability of AMOs as an ideal tool for midbrain-specific drug and toxicity screening with various high-throughput-compatible and functional readouts that can be directly applied to large-scale screening campaigns as well as more detailed secondary analyses to gain mechanistic insight into the behaviors of distinct cellular subpopulations in the context of their niche.

## Discussion

In this report, we present a fully scalable, automated workflow for the generation, maintenance, fixation, immunostaining, clearing, and optical high-content analysis of human midbrain organoids. In designing our workflow, we used a standard 96-well format along with optimized protocols providing full compatibility with standard liquid handlers in most screening facilities. By omitting cumbersome matrix embedding steps, we eradicate variation originating from positional differences of cells within extracellular matrix (ECM) droplets during embedding. Importantly, our workflow is also compatible with manual pipetting, and thus can benefit labs without access to liquid handlers. By starting with neural precursor cells rather than PSCs, we can accelerate and streamline the neural differentiation process resulting in highly reproducible, homogeneous, functionally coupled, and electrophysiologically active human 3D microtissues. In contrast to the highly organized layers of the cortex, which can be partially recapitulated in cerebral/cortical organoids (*Lancaster et al., 2013*; *Qian et al., 2016*; *Mariani et al., 2015*; *Bhaduri et al., 2020*), different tissues of the midbrain have a much less distinctive morphological organization at the cellular scale. While our AMOs do not represent the complexity of the actual human midbrain, we demonstrate the presence of key cell populations with a degree of structure and organization similar to that of other published midbrain organoids (*Qian et al., 2016*; *Jo et al., 2016*; *Monzel et al., 2017*). AMOs' uniformity in key parameters such as size, cellular organization, gene expression, and protein levels enables large-scale 3D-based HTS strategies. Moreover, the presence of aggregate-wide, reproducible, synchronous calcium and electrical activity in human neural 3D structures may enable neural activity to be used as a HTS-compatible, simple readout for phenotypic screens including toxicology studies and drug screening for disorders with altered brain activity.

While other groups have previously reported the use of human neural precursor cell (hNPC)-derived neurospheres for toxicity testing (*Moors et al., 2009*), as well as a hiPSC-derived high-throughput-compatible spheroid model including the use of calcium oscillations as a readout to evaluate neurotoxicity (*Sirenko et al., 2019*), these aggregates do not display structural tissue self-organization to the same extent as our organoids. Importantly, none of the available high-throughput-compatible protocols generates midbrain-specific organoids, but rather they focus on a cortical fate (*Sirenko et al., 2019*). Our model of the human midbrain opens up the potential to perform 3D organoid-based HTS of midbrain-specific disorders including the highly prevalent Parkinson's disease. One of the most widely used models for Parkinson's disease is the ablation of dopaminergic neurons with toxins such as 6-hydroxy dopamine or MPP+ (*Reinhardt et al., 2013b*; *Blum et al., 2001*; *Simola et al., 2007*; *Meredith and Rademacher, 2011*). Using these compounds as a dose-response model, our workflow demonstrates the unbiased, quantitative, cell-type-specific assessment of dopaminergic neurons in a physiologically relevant and complex human 3D model system with full scalability and high-throughput compatibility.

Our automated whole mount optical analysis can also be directly applied to other established 3D protocols and may help uncover phenotypes that manifest themselves primarily in 3D or in distinct subpopulations of cells. The neural precursor-based proof-of-principle demonstrated here may be

suitable, for example, in screens for diseases affecting primarily neural precursors including Zika virus infection. While this condition has been intensively studied using organoids, platform technologies for 3D high-throughput screens are still lacking and in high demand (*Qian et al., 2017*).

Furthermore, other non-midbrain organoid protocols may benefit from our workflow; for example, we have adapted cortical organoids by *Paşca et al., 2015*/*Sloan et al., 2018* to be scaled-up and analyzed in our system as automated hiPSC-based control organoids for our midbrain model. Automation of this existing manual protocol necessitated some modifications (summarized in *Figure 7—figure supplement 1*). For example, in order to create a workflow that is as scalable and as repeatable as possible, we decided to use accutase-digested single cell suspensions originating from feeder-free iPSC-cultures rather than dispase-treated colonies lifted off partially or whole from feeder-based iPSC-cultures. The use of colonies as a starting point makes colony/aggregate sizes difficult to control, especially in an automated setting, resulting in a variable starting point for each organoid generated in this fashion. We also aggregated and maintained the resulting organoids in SBS-compatible U-bottom plates that facilitate automated handling of all organoids as separate biological replicates, each in its own well, rather than in 100 mm dishes as in the original publication. These modifications to the original protocol may have changed the outcome compared to the published original; however, we demonstrate proper forebrain differentiation at the RNA and protein levels, and provide detailed single organoid-based high-content data from three independent batches for select parameters. We also believe that these modifications provide the most rigorous comparison between iPSC- and precursor-based organoid approaches by starting both protocols with well-defined and easily controllable cell numbers as single cell suspensions. By utilizing colonies as starting point, we would have biased reproducibility against the automated iPSC-protocol, thus weakening our comparison.

Despite our efforts to standardize iPSC-based organoids in a modified automated workflow, AMOs displayed a distinctly lower variance in a broad set of parameters, including size, cellular subpopulations, and survival in toxicity studies. This may indicate that mechanical standardization alone cannot compensate for the innate variability of PSCs during their self-organization.

Although the use of precursors cells as a starting population provides benefits, the comparison to hiPSC-based organoids also highlights certain limitations of this approach. Our choice of generating AMOs by seeding more committed neural precursor cells instead of hiPSCs improves the predictability of the differentiation outcome while maintaining the ability to form self-organized tissues. However, it also sacrifices some of the hiPSCs' broader cell fate potential and complexity resulting in simpler structures compared to hiPSC-based organoids. In addition, smNPCs can only give rise to cells of the ectoderm, which excludes, for example, the formation of microglia. Although smNPCs are capable of efficiently generating oligodendrocytes (*Reinhardt et al., 2013a*; *Ehrlich et al., 2017*), they generally arise very late in neural development (*Goldman and Kuypers, 2015*), and other organoid protocols also demonstrate the presence of oligodendrocytes after 100 days of differentiation or more (*Marton et al., 2019*). These long time scales are not very attractive for the design of high-throughput screening strategies. Instead, we favor approaches that ectopically add a known number of oligodendrocytes or microglia to our AMOs and thus reach a screenable state much earlier. Similar strategies have recently been described for the introduction of microglia into hiPSC-based neural organoids (*Lin et al., 2018*; *Abud et al., 2017*; *Muffat et al., 2018*), albeit without the ability to perform HTS. AMOs fill the gap between heterogeneous PSC-derived organoids and established but less physiological spheroid- and 2D-based screening formats.

The emerging science of 3D cell culture promises to probe the effects of drugs on single cell types (here: dopaminergic neurons) in an intact niche as part of a complex 3D in vitro culture. Understanding the effects in context will be an essential factor to better understand biology at the tissue level. Currently, 3D cell culture science is in a phase of transition, where traditional manual low throughput protocols need to be adapted to unbiased higher throughput workflows. This is a prerequisite to mine the information necessary to merge organoid science with the promises of big data and machine learning to better tackle the complexities of understanding 3D biology. Therefore, we see our workflow as a contribution to help others translate successful strategies such as cell painting (*Bray et al., 2016*) and other big-data-generating high-content strategies (*Friese et al., 2019*; *Scheeder et al., 2018*) into biologically relevant 3D models. If successful, this will help to open up single-cell-based phenotypic discovery to the third dimension by taking 3D-based HTS approaches beyond bulk techniques such as cell survival or gross morphology. Taken together, we

hope that our automation approach can contribute to establishing a next generation of cellular 3D in vitro disease models that allow unbiased, quantitative, high-throughput access to human tissue-surrogates in a dish.

# Materials and methods

Key resources table

| Reagent type (species) or resource | Designation | Source or reference | Identifiers | Additional information |
|---|---|---|---|---|
| Cell line (*Homo sapiens*) | AMO line 1 | *Reinhardt et al., 2013b* | PMID:23533608 | smNPCs used for the derivation of AMOs designated 'AMO line 1' |
| Cell line (*Homo sapiens*) | AMO line 2 | *Reinhardt et al., 2013a* | PMID:23533608 | smNPCs used for the derivation of AMOs designated 'AMO line 2' |
| Cell line (*Homo sapiens*) | hIPSCs | *Reinhardt et al., 2013a* | PMID:23472874; PMID:23533608 | hIPSCs giving rise to hIPSC organoids in this paper; cell line of origin to generate smNPCs for AMO line 2 |
| Antibody | Anti-Brn2 (Rabbit monoclonal) | Cell Signaling | Cat#:12137 | (1:2000) |
| Antibody | Anti-Cleaved Caspase-3 (Rabbit monoclonal) | Cell Signaling | Cat#:9664 | (1:100) |
| Antibody | Anti-Ctip2 (Rat monoclonal) | Abcam | Cat#:ab18465 | (1:750) |
| Antibody | Anti-DCX (Goat polyclonal) | Santa Cruz | Cat#:sc-8066 | (1:500) |
| Antibody | Anti-FoxA2 (Mouse monoclonal) | Santa Cruz | Cat#:sc-101060 | (1:100) |
| Antibody | Anti-FoxG1 (Rabbit polyclonal) | Abcam | Cat#:ab18259 | (1:500) |
| Antibody | Anti-GFAP (Chicken polyclonal) | Merck Millipore | Cat#:AB5541 | (1:500) |
| Antibody | Anti-Lmx1a (Rabbit polyclonal) | Abcam | Cat#:ab139726 | (1:100) |
| Antibody | Anti-Homer (Mouse monoclonal) | Synaptic Systems | Cat#:160 011 | (1:250) |
| Antibody | Anti-Map2 (Chicken polyclonal) | Abcam | Cat#:ab5392 | (1:500) |
| Antibody | Anti-Map2 (Mouse monoclonal) | Merck Millipore | Cat#:MAB3418 | (1:1000) |
| Antibody | Anti-Map2 (Rabbit polyclonal) | Abcam | Cat#:ab32454 | (1:500) |

*Continued on next page*

*Continued*

| Reagent type (species) or resource | Designation | Source or reference | Identifiers | Additional information |
|---|---|---|---|---|
| Antibody | Anti-Nestin (Mouse monoclonal) | Life Technologies | Cat#:MA1-110 | (1:250) |
| Antibody | Anti-Nurr1 (Mouse monoclonal) | Santa Cruz | Cat#:sc-376984 | (1:100) |
| Antibody | Anti-Pax6 (Rabbit polyclonal) | BioLegend | Cat#:901301 | (1:500) |
| Antibody | Anti-Pitx3 (Rabbit polyclonal) | Merck Millipore | Cat#:AB5722 | (1:100) |
| Antibody | Anti-S100b (Rabbit polyclonal) | Dako | Cat#:Z031129-2 | (1:500) |
| Antibody | Anti-Satb2 (Mouse monoclonal) | Abcam | Cat#:ab51502 | (1:500) |
| Antibody | Anti-Sox2 (Goat polyclonal) | R and D Systems | Cat#:AF2018 | (1:200) |
| Antibody | Anti-Synapsin1 (Mouse monoclonal) | Synaptic Systems | Cat#:106 001 | (1:1000) |
| Atibody | Anti-Synaptophysin1 (Rabbit polyclonal) | Synaptic Systems | Cat#:101 002 | (1:200) |
| Antibody | Anti-Tbr1 (Rabbit polyclonal) | Abcam | Cat#:ab31940 | (1:500) |
| Antibody | Anti-Tbr2 (Chicken polyclonal) | Merck Millipore | Cat#:AB15894 | (1:500) |
| Antibody | Anti-TUBB3 (Mouse monoclonal) | BioLegend | Cat#:801202 | (1:500) |
| Antibody | Anti-TH (Chicken polyclonal) | Abcam | Cat#:ab76442 | (1:1000) |
| Antibody | Anti-TH (Rabbit polyclonal) | Abcam | Cat#:ab112 | (1:500) |
| Antibody | Anti-vGAT (Mouse monoclonal) | Synaptic Systems | Cat#:131 011 | (1:100) |
| Antibody | Anti-vGLUT1 (Rabbit polyclonal) | Synaptic Systems | Cat#:135 303 | (1:100) |
| Sequence-based reagent | AADC_F | This paper | PCR primers | TGCGAGCAGAGAGGGAGTAG |
| Sequence-based reagent | AADC_R | This paper | PCR primers | TGAGTTCCATGAAGGCAGGATG |

*Continued on next page*

*Continued*

| Reagent type (species) or resource | Designation | Source or reference | Identifiers | Additional information |
|---|---|---|---|---|
| Sequence-based reagent | Brn2_F | This paper | PCR primers | CGGCGGATCA AACTGGGATTT |
| Sequence-based reagent | Brn2_R | This paper | PCR primers | TTGCGCTGCG ATCTTGTCTAT |
| Sequence-based reagent | DCX_F | This paper | PCR primers | AGGGCTTTCTT GGGTCAGAGG |
| Sequence-based reagent | DCX_R | This paper | PCR primers | GCTGCGAATCT TCAGCACTCA |
| Sequence-based reagent | EN1_F | This paper | PCR primers | CCCTGGTT TCTCTGGGACTT |
| Sequence-based reagent | EN1_R | This paper | PCR primers | GCAGTCTGTGG GGTCGTATT |
| Sequence-based reagent | GAPDH_F | This paper | PCR primers | CTGGTAAAGTG GATATTGTTGCCAT |
| Sequence-based reagent | GAPDH_R | This paper | PCR primers | TGGAATCATATT GGAACATGTAAACC |
| Commercial assay or kit | Biomark 48.48 integrated fluidic circuit Delta Gene assay | Fluidigm | Cat#:101–0348 | Complete bundle for 10 assays |
| Commercial assay or kit | CellTiter-Glo 3D Cell Viability Assay | Promega | Cat#:G9682 | |
| Commercial assay or kit | Dopamine ELISA Kit | Abnova | Cat#:KA3838 | |
| Commercial assay or kit | CellTracker deep red dye | Life Technologies | Cat#:C34565 | |
| Commercial assay or kit | Fluo-4 AM | Thermo Fisher | Cat#:F14201 | |
| Chemical compound, drug | Cobalt(II) chloride | Sigma-Aldrich | Cat#:232696 | |
| Chemical compound, drug | G418 | Sigma-Aldrich | Cat#:G8168 | |
| Chemical compound, drug | 6-Hydroxydopamine hydrochloride (6OHD) | Sigma-Aldrich | Cat#:H4381 | |
| Chemical compound, drug | 1-Methyl-4-phenylpyridinium iodide (MPP+) | Sigma-Aldrich | Cat#:D048 | |
| Chemical compound, drug | Dopamine hydrochloride | Sigma-Aldrich | Cat#:H8502 | |
| Chemical compound, drug | Risperidone | Sigma-Aldrich | Cat#:R3030 | |

*Continued on next page*

*Continued*

| Reagent type (species) or resource | Designation | Source or reference | Identifiers | Additional information |
|---|---|---|---|---|
| Chemical compound, drug | GABA | Sigma-Aldrich | Cat#:A2129 | |
| Chemical compound, drug | Bicuculline | Sigma-Aldrich | Cat#:14340 | |
| Chemical compound, drug | Glutamate | Sigma-Aldrich | Cat#:49621 | |
| Chemical compound, drug | Glycine | Sigma-Aldrich | Cat#:50046 | |
| Chemical compound, drug | Ketamine | Sigma-Aldrich | Cat#:K2753 | |
| Software, algorithm | Fiji | *Schindelin et al., 2012* | PMID:22743772 | |
| Software, algorithm | GraphPad Prism | Graphpad Software Inc | RRID:SCR_002798 | |
| Software, algorithm | Harmony | Perkin Elmer | Version:'4.1, Revision 128972' | |
| Software, algorithm | Columbus | Perkin Elmer | Version:2.6.0. 127073 | |

## smNPC culture

All cell lines used in this study tested negative for mycoplasma contamination in PCR- and sequencing-based analyses. Unless otherwise noted, all cells and 3D aggregates were maintained at 37°C and 5% $CO_2$. The human small molecule precursor cells (smNPCs) were generated and characterized during a previous study (*Reinhardt et al., 2013a*) and cultured as described there, with minor modifications. We grew smNPCs in 1% (v/v) Matrigel (BD)-coated 6-well plates (Sarstedt) in N2B27 medium supplemented with the small molecules smoothened agonist (SAG) (0.5 µM, Cayman Chemical) and CHIR 99021 (3 µM, Axon MedChem). N2B27 consisted of DMEM-F12 (Thermo Fisher) and Neurobasal Medium (Thermo Fisher) at a 1:1 ratio, enriched with 1:400 diluted N2 supplement (Thermo Fisher), and 1:200 diluted B27 supplement without vitamin A (Thermo Fisher), 1% penicillin/ streptomycin/glutamine (Thermo Fisher), and 200 µM ascorbic acid (Sigma-Aldrich). Typically, we exchanged medium every other day. The cells were split every 5–7 days at a splitting ratio of 1:10 to 1:20 via accutase (Sigma-Aldrich) treatment for approximately 15 min at 37°C, yielding a single-cell solution. To stop the digestion, the cells were diluted in DMEM-F12 with 0.1% bovine serum albumin (BSA) (Thermo Fisher) and centrifuged at 220 g for 2 min. The cell pellet was resuspended in fresh smNPC medium (N2B27 with 0.5 µM SAG and3 µM CHIR) and plated on Matrigel-coated 6-well plates.

## Definition of batch

Each batch consisted of independently frozen and thawed cells of the same passage and cell line, passaged separately and seeded, maintained, fixed, stained, and cleared separately from the other batches.

## AMO generation

All liquid handling steps (seeding, maintenance, and fixation of organoids) were performed by a Beckman Coulter Biomek FXP liquid handling station equipped with a 96-channel-pipetting head and an attached Cytomat incubator (Thermo Fisher). After digestion by accutase, we seeded 9000 smNPCs in each well of a conical 96-well plate (Thermo Fisher) in smNPC medium and allowed them

to aggregate for 2 days. To increase inter-cell adhesion, we added 0.4% (w/v) polyvinyl alcohol (PVA, Sigma-Aldrich). Starting at day 2, cells undergo ventral patterning over 4 days in two feedings by removal of CHIR99021 in the continued presence of 1 µM SAG and the addition of 1 ng/mL brain derived neurotrophic factor (BDNF, PeproTech) and 1 ng/mL glial cell line-derived neurotrophic factor (GDNF, PeproTech). After ventralization, we removed SAG on day 6, further supported midbrain differentiation and maturation by the addition of 1 ng/mL transforming growth factor beta 3 (TGFβ−3, PeproTech), and 100 µM dibutyryl cyclic adenosine monophosphate (dbcAMP, Sigma-Aldrich). To boost maturation and cell survival during the rest of the neural maturation, we increased the concentration of BDNF and GDNF to 2 ng/mL each starting at day 6. A single dose of 5 ng/mL Activin A (eBioscience) was added on day 6 only. Depending on the desired degree of maturity, the duration of the maturation phase can be prolonged to 100 days and longer. Organoids were fed every other day for the entire duration of culture.

## Size measurement of organoids

For size measurements, we took brightfield images of randomly selected AMOs or hiPSC organoids using a stereo microscope (Leica MZ10 F, camera: Leica DFC425 C). Images were processed with ImageJ/Fiji (*Schindelin et al., 2012*) using a custom-tailored standardized workflow. The auto threshold function was used to discriminate samples from the background followed by a measurement of their area with the analyze particles function. The measured area corresponds to the largest cross-section of the organoid. Data were outputted to Microsoft Excel and GraphPad Prism v8.4.2 (Graphpad Software, Inc) for further analysis. Coefficients of variation (CVs) were calculated via CV = standard deviation/mean.

## Whole mount staining and clearing

In order to analyze protein expression in 3D in a HTS-compatible manner, we adapted a whole mount staining protocol based on *Lee et al., 2016*. for large-scale 3D aggregates and optimized it for use in an automated liquid handling system (Beckman Coulter Biomek FXP, 96-channel-pipetting head). After fixation with 4% PFA (VWR) for 10–15 min, we stained the aggregates with primary and secondary antibodies (Alexa Fluor secondary antibodies, Thermo Fisher) for 6 days each. A list of all primary antibodies and concentrations can be found in *Supplementary file 3*. We diluted the antibodies in a blocking and permeabilization solution (6% BSA, 0.5% Triton-X 100 (Roth), 0.1% (w/v) sodium azide (Sigma-Aldrich) in PBS (Sigma-Aldrich)) and renewed it every 2 days. Between primary and secondary antibody incubation as well as after the staining procedure we washed the samples 5 times for 1 hr with 0.1% Triton X-100 in PBS. This extremely long staining procedure allows the antibodies to fully penetrate the aggregates despite their large size and high density. To enable full penetration by microscope illumination, the whole mount staining procedure is followed by BABB-based tissue clearing (*Dent et al., 1989*). First, the samples were dehydrated stepwise through a methanol (Roth) series (25%, 50%, 70%, 90%, 100%, 15 min each). Next, they were transferred to an organic solvent-resistant cyclo-olefin 96-well plate ('Screenstar', Greiner Bio-One). The samples were incubated for 30 min in 1:1 methanol/BABB (benzyl benzoate (Sigma-Aldrich) and benzyl alcohol (Sigma-Aldrich) 1:1) and subsequently kept in BABB for imaging. We used Imaris v9.1.2 (Bitplane, Oxford Instruments) for 3D rendering of confocal slices to produce *Video 1*.

## Quantitative real-time PCR

We performed RNA isolation for quantitative real-time PCR (qPCR) analysis using the NucleoSpin RNA XS kit (Macherey-Nagel) according to the manufacturer's instructions. Depending on the age and protocol used, we pooled organoids from one batch in order to yield enough RNA for downstream analysis; AMOs: 32 (d6), 24 (d16), or 18 (d22 and d30), hiPSC organoids: 32 (d6), 14 (d22), or 10 (d30). We determined RNA concentration and purity using a NanoDrop 8000 spectrophotometer (Thermo Fisher) and performed reverse transcription according to standard protocols using 1000 ng RNA per reaction. qPCR was done in triplicates on a Quantstudio 5 Real-Time PCR System (Applied Biosystems) with iTaq Universal SYBR Green Supermix (Bio-Rad) and 3.2 ng RNA equivalents per 10 µL reaction. Cycling conditions were 2 min at 95℃ followed by 40 cycles of 15 s at 95℃ and 60 s at 60℃. We calculated the relative expression using the ΔΔct method and normalized to undifferentiated smNPCs (AMO line 1) collected before aggregation as well as using GAPDH as a housekeeping

gene. Alternatively, gene expression was quantified with the Biomark 48.48 integrated fluidic circuit (IFC) Delta Gene assay (Fluidigm) according to the manufacturer's instructions. Briefly, following 14 cycles of preamplification, the samples were subjected to an exonuclease I (New England Biolabs) treatment (37˚C for 30 min and 80˚C for 15 min) and diluted twentyfold with DNA Suspension buffer (TEKnova). The samples (in duplicates) and assay mixtures were loaded onto a 48.48 microfluidic IFC chip and run on the BioMark real-time PCR reader (Fluidigm) where they were amplified and measured according to manufacturer's instructions. Here, data analysis was performed using the BioMark real-time PCR analysis software 4.3.1 (Fluidigm) with standard settings. Again, undifferentiated smNPCs (AMO line 1) were used as a reference and GAPDH served as housekeeping gene. All data was transferred to Microsoft Excel for further processing and GraphPad Prism v8.4.2 for plotting. A list of all used primers can be found in *Supplementary file 4*.

## Calcium imaging

For calcium imaging, we added 10 µM cell-permeant Fluo-4 AM (Thermo Fisher) diluted in AMO medium to the aggregates and incubated for 60 min at 37˚C. For inhibitor studies cobalt(II) chloride (Sigma-Aldrich) was added to the medium at a concentration of 2 mM together with the Fluo-4 AM. Imaging was performed using a Dragonfly spinning disc confocal microscope (Andor, Oxford Instruments) at a frequency of 10 Hz for 4 min. Data analysis was performed using ImageJ/Fiji (*Schindelin et al., 2012*). First, different ROIs were defined as depicted in *Figure 4*. Then, the mean fluorescence intensity in those ROIs was measured over time and plotted using GraphPad Prism v8.0.2. The video was assembled via ImageJ/Fiji (*Schindelin et al., 2012*) and the frame rate accelerated to compress 4 min real time at 10 Hz into 20 s running time. Alternatively, we measured fluorescence intensity on a Synergy Mx plate reader (BioTek), acquired data with the Gen5 software (BioTek) and outputted it to Microsoft Excel and GraphPad Prism v8.0.2 for further analysis and plotting.

## Electrophysiological analysis of AMOs by microelectrode array

Electrophysiological measurements on microelectrode arrays (USB-MEA256system, Multichannel Systems) were performed on electrode areas of 9-well MEAs as previously described (*Piccini et al., 2017*). The MEAs were plasma-cleaned and coated with 1:75 diluted Matrigel (Corning) in KO-DMEM (Invitrogen) overnight and additionally for 2 hr before seeding with a 0.1% gelatin (Sigma-Aldrich) in PBS solution at room temperature. The MEAs were pre-warmed to 37˚C, organoids were transferred to the electrode area of the MEAs, and allowed to attach for approximately 28 hr. Subsequently, MEA chambers with attached samples were used for electrophysiological recordings at 37˚C. To study the effects of different pharmacological modulators, organoids were first measured under basal conditions (i.e. no addition of compounds) to record a reference signal. We then added specific pathway activators to each sample chamber of a MEA and recorded the electric field potential after a brief period of equilibration. After recording the signal for the activators, we next added the inhibitors for the respective pathways to the same well and repeated the measurement procedure as before. This guaranteed that we measured the change of electric field potential of each sample and the basal activity of each sample could serve as an internal control. The compounds used were dopamine hydrochloride (10 µm, Sigma-Aldrich), risperidone (10 nM, Sigma-Aldrich), GABA (10 µm, Sigma-Aldrich), bicuculline (1 µm, Sigma-Aldrich), glutamate/glycine (100 µm each, Sigma-Aldrich), and ketamine (10 µm, Sigma-Aldrich). It is possible to culture organoids on MEAs for longer and to achieve a more widespread contact with MEA electrodes. However, organoids flatten and change morphology and possibly cellular composition upon prolonged attachment, and we sought to measure the organoids' electrical activity as close to their usual spherical state as possible. 28 hr allowed sufficient time for attachment of the aggregates to record electric fields from still spherical samples while providing enough mechanical cell-substrate connection to transport MEA substrates from the incubator to the recording rig. After the recordings, it was possible to gently remove the organoids without causing observable damage. Although not optimized (as we only performed measurements at one time point per organoid), this opens up the possibility to transfer the organoids back to standard culture conditions after MEA measurements and thus perform multiple measurements over time. Datasets were recorded with Cardio2D software (Multichannel Systems). Analyses were performed using the software Cadio2D+ (Multichannel Systems) and Origin v9.0

(OriginLab) on exported data. Discrete fast Fourier analyses in Origin (Blackman window) was used to assess frequencies of autonomous activity of the organoids. To compare neural sample activity via electric field oscillations (*Figure 8k*), we pre-processed the raw MEA data with a Savitzky-Golay filter (curves were smoothed with a window size of 50 to preserve peak data while removing noise) in Origin v9.0 and formed the total sum of the absolute values of each oscillation over 15 s. The results were outputted to Microsoft Excel, reformatted and then transferred to GraphPad Prism v8.4.2 for plotting.

## Electrophysiological analysis of single cells by patch-clamping

Due to the morphology of AMOs (high optical density and the fact that most cell bodies are located in a depth of at least 10–20 µm), it was technically impossible to perform the patch-clamp measurements on intact aggregates. Therefore, the organoids were treated with 1 mg/ml trypsin and then mechanically dispersed to obtain single cells. These were seeded on PDL-coated coverslips and cultured for 1–3 days in AMO medium (we stated the age of AMOs at the time of dissociation). The transmembrane currents were recorded from isolated cells using the whole-cell configuration of the patch-clamp technique (*Hamill et al., 1981*). The patch pipettes were fabricated from borosilicate glass on a Sutter P1000 (Sutter Instrument company) pipette puller. When filled with pipette solution, they had a tip resistance of 4–6 MΩ. Recordings were done using an EPC-10 amplifier (HEKA Elektronik) and Patchmaster acquisition software (HEKA Elektronik). Series resistance, liquid junction potential, pipette and whole-cell capacitance were cancelled electronically. Bath solution contained (mM): NaCl 140, KCl 2.4, MgCl2 1.2, CaCl2 2.5, HEPES 10, D-glucose 10, pH 7.4 and the pipette solution contained (mM): K-aspartate 125, NaCl 10, EGTA 1, MgATP 4, HEPES 10, D-glucose 10, pH 7.4 (KOH). We performed all experiments at room temperature. Recordings of current-voltage relationship (I-V curves) were done in voltage-clamp mode at a holding potential of −70 mV. Recordings of evoked action potentials were performed in current-clamp mode. Data were analyzed using Patcher's Power Tool routine for IgorPro (WaveMetrics), SciDAVis (http://scidavis.sourceforge.net/) and Origin Pro 2019 (Origin Lab). To reveal the shape of I-V curves, single traces were normalized to the peak amplitude and then averaged.

## RNA sequencing

To isolate RNA of single AMOs and organoids we used the Direct-zol-96 RNA kit (Zymo Research) according to the manufacturer's instructions. We assessed RNA concentration and purity using a NanoDrop 8000 spectrophotometer and RNA integrity with a Bioanalyzer (Agilent Technologies) per standard protocols. Next, mRNA was enriched using the NEBNext Poly(A) Magnetic Isolation Module (NEB) followed by strand-specific cDNA NGS library preparation (NEBNext Ultra II Directional RNA Library Prep Kit for Illumina, NEB). The size of the resulting library was controlled by use of a D1000 ScreenTape (Agilent 2200 TapeStation) und quantified using the NEBNext Library Quant Kit for Illumina (NEB). Equimolar pooled libraries were sequenced in a single read mode (75 cycles) on the NextSeq 500 System (Illumina) using v2 chemistry yielding in an average QScore distribution of 95% >= Q30 score and subsequent demultiplexed and converted to FASTQ files by means of bcl2fastq v2.20 Conversion software (Illumina).

## RNA sequencing analysis

We aligned the RNA sequencing reads to the human genome hg19 with TopHat2 aligner (v2.1.1) (*Kim et al., 2013*), using default input parameters. Gene annotation from Ensembl (version GRCh37.87) were used in the mapping process. The number of reads that were mapped to each gene was counted using the Python package HTSeq (v0.7.2) (*Anders et al., 2015*) with 'htseq-count – mode union – stranded no'. For the correlation analysis, sequencing data of 21 different human fetal organs and midbrain(-like) samples were obtained from GSE66302 and E-MTAB-4868, respectively. Human prenatal cortex (at 24 weeks post-conception) RNA-seq datasets were obtained from www.nature.com/neuro/journal/v18/n1/extref/nn.3898-S9.zip. Reads were mapped to the human genome as described above. RPKM values (Reads Per Kilobase of transcript per Million mapped reads) for each gene were computed by Cufflinks (v2.2.1). We selected genes with high expression (log RPKM > 1) for further analysis. Based on the expression of the selected genes, Pearson correlations were calculated. We averaged the correlation coefficients for biological replicates. PCA and

differential expression analysis were performed with raw counts using the R package DESeq2 (v1.18.1). Genes were considered as deregulated if |log2FC| > 2 and FDR < 0.05 using Benjamini-Hochberge multiple test adjustment (*Benjamini and Hochberg, 1995*). Gene Ontology (GO) term enrichment was analyzed with the bioinformatics web server Gorilla (*Eden et al., 2009*) and visualized with REViGO (*Supek et al., 2011*).

For the comparison with cerebral organoids (*Figure 5—figure supplement 1*), the dispersion within groups (*Figure 5—figure supplement 1b*) was calculated using the average distance between data points and centroids on the PCA plot (*Figure 5—figure supplement 1a*).

All RNA sequencing data generated by us was deposited to the NCBI GEO database (GSE119060) and can be accessed at https://www.ncbi.nlm.nih.gov/geo/query/acc.cgi?acc= GSE119060.

## Quantification of whole mount staining and clearing

To assess how quantitative our imaging workflow is, we performed a dilution experiment. We mixed unlabeled smNPCs with different percentages (1.25%, 2.5%, 5%, 10%, 20%, 40%) of CellTracker deep red dye (Life Technologies)-labeled cells (labeling according to standard protocols, dye concentration 1:20,000) and aggregated them in smNPC maintenance medium with 0.4% PVA. To explore the effects of overall aggregate size on quantitation, we generated aggregates with 100,000 as well as 200,000 cells in total. After 1 day of aggregation, the aggregates were fixed with 4% PFA, subjected to BABB-based tissue clearing, imaged, and analyzed as described below.

## General workflow for high-content imaging and analysis

After staining and clearing, we achieved uniform aggregate positioning within the wells by tilting the plates off the horizontal plane at 60 degrees for 1 min. Image acquisition was carried out in an Operetta high content imager (Perkin Elmer) and images were analyzed in Harmony 4.1 software and/or Columbus version 2.6.0.

Each high-content imaging experiment requires a customized set of parameters adjusted for the size of the aggregates, marker-wavelengths, their morphology and distribution within the sample, marker combinations, and signal and background intensities. As the details of each workflow are particular to our hardware and software setup, we first provide a general description of our workflow that focuses on explaining the principal steps in a manner that allows platform-independent reproduction. We then provide the detailed analysis pipeline for the data shown in *Figure 6* (quantification of Sox2 and Map2 in AMO line 1) with all its unique parameters and steps in the Materials and methods section titled 'Detailed high-content imaging analysis example for the data shown in *Figure 6*'.

### Acquisition

As our version of high content imager did not include automatic sample-finding capability, we started by acquiring well overview images at 2x magnification. Although uniform, locations of aggregates were not identical from well to well, necessitating acquisition of tiled 3D stacks of adjacent fields of view (FOV) with 10x magnification covering all possible locations of aggregates within a small area of given wells. The 2x overview images guided the choice of location for the 10x FOVs. This resulted in tiled non-overlapping stacks of images, some of which contained aggregates, and some of which did not. These 10x images entered further image analysis.

### Analysis

In general, after hardware-based background correction, image analysis was performed separately for each FOV and confocal plane, and results were either reported as mean or sum for all FOVs and planes per well as indicated in the figures and figure legends. Further analysis consisted of three principal stages: (1) sample identification, (2) background correction, (3) segmentation, and (4) data reporting.

1. To identify aggregates and distinguish them from occasional contaminations, we summed all acquired channels (or in case of high background noise in particular channels used a single one with homogeneous signal) and smoothed the resulting image with either a gaussian or a median kernel to remove local outlier pixels. Simple thresholding identified organoids as

regions of interest (ROI), which were further refined via morphology (e.g. area and roundness) and intensity to exclude incorrectly identified objects such as dust or fibers.

2. As large spherical aggregates possessed a non-uniform background signal, we performed local background correction before image segmentation. Subtracting a blurred version of the image from the raw image removed most of the non-uniform haze in each plane. To boost fine structures, sliding-parabola algorithms provided additional local contrast when needed.

3. Structures of interest within each bona fide aggregate identified in (1), e.g. nuclei or cells, were segmented by either simple thresholds or by dedicated proprietary algorithms (often called 'find nuclei' or 'find cells') and further refined by morphology and intensity if needed.

4. Finally, the parameters of interest (often intensity, area, number of objects) were measured from the raw image channel based on the previously defined and selected ROIs. If object segmentation was not possible (e.g. due to insufficient resolving power at 10x for highly filamentous markers or low signal to noise ratios), raw/total integrated signal intensity within the defined sample region was reported. In order to generate results for the entire 3D aggregate, which may span several FOVs, datasets for all planes and fields were combined. Often, it is also necessary to normalize the data to the aggregate area to account for sample variations across planes and wells.

## Detailed high-content imaging analysis example for the data shown in *Figure 6*

After staining and clearing, we achieved uniform aggregate positioning within the wells by tilting the plates off the horizontal plane at 60° for 1 min. Image acquisition was carried out in an Operetta high content imager (Perkin Elmer) and images were analyzed in Harmony 4.1 software. We acquired a total of 16 confocal planes in three channels (DAPI, Sox2 488 nm, and MAP2 647 nm) with an inter-plane spacing of 36.6 µm for a total stack of 549 µm, covering the entire aggregate height (clearing and dehydration steps tend to shrink and flatten aggregates slightly). To define the aggregate region on each image plane, all three channels were summed, filtered with a median filter to remove small localized features, and bright areas were identified via the 'find image region' function. After cleaning the edge of the aggregate region by dilation and erosion steps of 10 and 3 pixels, respectively, we identified bona fide AMOs by selecting for regions with a minimum of 300 arbitrary brightness units (abu) and 4000 µm$^2$ size. In order to better isolate Sox2+ nuclei from the general background, we ran a sliding parabola algorithm with a curvature setting of 2 across each image plane in the 488 nm channel. Nuclei were then identified within each aggregate region via the 'find nuclei' function, algorithm 'M' and further selected to be Sox2+ if they were larger than 10 µm$^2$ and brighter than 1200 abu. We excluded image artifacts, small dust particles, and overlapping nuclei by omitting nuclei brighter than 6000 abu and larger than 70 µm$^2$ from further quantification. For final output, the number and total brightness of nuclei in 488 nm and of aggregate regions in 647 nm were summed for all planes and all fields of view in each well and transferred to Microsoft Excel and TIBCO Spotfire for further annotation, analysis, and plotting. We omitted data from wells that contained dust particles within the same FOV as organoids, incompletely imaged aggregates due to improper positioning, or AMOs that have been damaged or lost during culture or downstream processing. Plate 1, 2, and 3 represent independent batches of separately thawed and cultured cells of the same frozen batch.

## Human-induced pluripotent stem cell culture

For cortical organoid generation (see method section below), human induced Pluripotent Stem Cells (hiPSCs) (parental line of AMO line 2 smNPCs) were generated and characterized during a previous study (*Reinhardt et al., 2013b*). Cells were cultured in Vitronectin (Thermo Fisher)-coated 6-well plates in mTeSR Plus medium (Stemcell technologies) supplemented with 1% penicillin/streptomycin. The medium was changed every other day and cells were passaged at a ratio of 1:10 to 1:15, using accutase when they reached 80–90% confluency. After splitting, the medium was supplemented with the ROCK inhibitor Y-27632 (10 µM, tebu-bio) until the first media exchange.

For the generation of iPSC-based cerebral organoids (see method section below and *Figure 5— figure supplement 1*) hiPSC culture was performed feeder-free using modified FTDA medium (*Frank et al., 2012*) in 1% (v/v) Matrigel-coated 6-well plates with iPSCs from the same line as AMO line 2. FTDA medium consisted of DMEM-F12 supplemented with 1% human serum albumin

(Biological Industries), 1% Chemically Defined Lipid Concentrate (Life Technologies), 0.1% Insulin-Transferrin-Selenium (BD), 1% penicillin/streptomycin/glutamine. We fed the iPSCs daily and added 10 ng/mL FGF2 (PeproTech GmbH), 0.2 ng/mL TGFβ3 (PeproTech GmbH), 50 nM Dorsomorphin (Enzo Life Sciences), 5 ng/mL Activin A (eBioscience), 20 nM C59 (Tocris) before each media exchange. We split the iPSCs as single cells every 3–5 days using accutase for approximately 10 min at 37°C. We transferred 600,000 cells per well of a 6-well plate to be seeded to DMEM-F12 with 0.1% BSA and centrifuged at 220 g for 2 min. We resuspended the cell pellet in fresh FTDA medium supplemented with 1:2000 ROCK inhibitor Y-27632 (tebu-bio) and plated the iPSCs on Matrigel-coated 6-well plates.

## Generation of automated cortical organoids

As a control for our AMOs, we generated cortical hiPSC organoids from the same hiPSC line used to derive smNPCs for AMO line 2 (*Reinhardt et al., 2013b*). After manual 2D culture of hiPCs, all steps were fully automated using our liquid handling system (Beckman Coulter) with attached incubator (Thermo Fisher). Generally, we followed the protocol previously published by *Paşca et al., 2015*. (and also described in more detail by *Sloan et al., 2018*), with adaptations for our automation pipeline (see *Figure 7—figure supplement 1*). Starting with 90–100% confluent cultures, we detached hiPSCs with accutase and seeded 10,000 cells per well in ultra-low attachment U-bottom plates (Corning). Cortical organoid medium consisted of DMEM F-12, 20% Knock-out Serum replacement (GIBCO), 1% penicillin/streptomycin/glutamine, 1% Non-essential amino acids (Sigma-Aldrich), and 0.2% 2-Mercaptoethanol (Thermo Fisher). For the first 6 days, we supplemented the cortical organoid medium with 5 µM dorsomorphin (Enzo Life Sciences) and 10 µM SB-431542 (Biomol). During seeding only, we also added 10 µM ROCK inhibitor Y-27632. Aggregates were fed every 3 days using an automated liquid handling system. From day 6 to 24, culture medium supplements were exchanged to EGF and FGF2 (both 20 ng/ml, PeproTech) and afterwards BDNF and NT3 (metabion) (both 20 ng/ml).

## Organoid viability assay

To measure the viability of individual organoids we used the CellTiter-Glo 3D Cell Viability Assay (Promega) according to the manufacturer's instructions. The entire procedure was performed using an automated liquid handling system (Beckman Coulter) and is thus fully scalable and HTS-compatible. In short, the reagent and the AMOs were brought to room temperature in their 96-well culture plates for 30 min and the media volume of each 96-well was adjusted to 55 µl. We added an equal volume (55 µl) of the CellTiter Glo 3D reagent and let it shake on a Thermomixer (Eppendorf) at 900 rpm for 5 min before incubating the samples protected from light at room temperature for 25 min. To prevent cross-talk between wells when measuring the luminescence, we next transferred the contents from the clear 96-well culture plates to opaque white 384-well Lumitrac plates (Greiner) with two technical replicates per sample. Luminescence was recorded immediately after transfer with a Synergy Mx plate reader (BioTek). The results were outputted to Microsoft Excel, reformatted and then transferred to GraphPad Prism v8.4.2 for plotting. Coefficients of variation (CVs) were calculated via CV = standard deviation/mean.

## Dopaminergic neuron-specific toxicity testing and quantification

On day 47 of differentiation, AMOs received Tox medium (TM) overnight to remove antioxidants present in the B27 media supplement. TM consisted of DMEM-F12 supplemented with 1% N2 and 1% penicillin/streptomycin/glutamine. The next day, AMOs were treated with different concentrations (0, 50, 100, 250, 500 µm) of either 6-Hydroxydopamine hydrochloride (Sigma-Aldrich) or 1-Methyl-4-phenylpyridinium iodide (MPP+, Sigma-Aldrich) in TM. To refresh compounds, AMOs received TM including toxins at the same concentrations as before 24 hr after the first application. After a total incubation of 48 hr, the medium was changed back to standard AMO medium, and the samples were cultured for 6 more days to allow cell death to occur and the dead cells to be cleared from the organoids. After 6 days, the samples were fixed with 4% PFA, whole mount stained for Map2 and TH, BABB-cleared, and subjected to high-content confocal imaging as described above. We performed image analysis following the steps as outlined in the high-content analysis section with slight modifications to accommodate the individual brightness, morphology, and background

characteristics of the staining. Briefly, after identifying AMOs, we ran a Gaussian smoothing algorithm across the TH 647 nm (10 px width) and Map2 488 nm (five px width) channels and subtracted the smoothed images from the raw images to better isolate TH+ and Map2+ cells from the background. In the TH 647 nm channel, cells were then identified with the 'find cells' function, algorithm 'C' and further selected to be TH+ if they were brighter than 200 abu and larger than 25 $\mu m^2$. For Map2, an additional sliding parabola algorithm with a curvature of 10 was run across the subtracted image to further reduce background noise. Map2+ cells were then also identified using the 'find cells' function algorithm 'C'. As final output, the total intensity of the identified TH+ cells in the raw 647 nm channel and the Map2+ cells in the raw 488 nm channel were summed for all fields of view and confocal planes per well. The data was then transferred to Microsoft Excel for further analysis, including normalization to the organoid area, and plotted using GraphPad Prism v8.4.2.

## Dopamine secretion

We collected the cell culture supernatant from 35 days old organoids 40 hr after feeding and measured its dopamine content using a Dopamine ELISA Kit (Abnova) according to the manufacturer's instructions. Measurements were performed in duplicates and the sample concentrations were calculated based on a standard curve.

## Determining the proportion of mature cells

Since there is no single marker that identifies all different mature cell populations within the AMOs, but all immature smNPCs express the precursor marker Sox2, which is downregulated upon maturation, we defined Sox2-negative cells as mature for the purpose of this analysis. Using confocal high-content imaging analysis as described above, we quantified the number of Sox2-positive cells as well as the total cell number (based on DAPI-stained nuclei) within the entire organoids. We then calculated the ratio of Sox2-negative cells as 1 - (number of Sox2-positive cells) / (total cell number). Calculations were performed in Microsoft Excel and data was transferred to GraphPad Prism v8.4.2 for plotting.

## ScaleSQ clearing

ScalesSQ tissue clearing was performed as previously described by *Hama et al., 2015*. Briefly, we incubated organoids in ScaleSQ (22.5% (w/v)) D-sorbitol (Sigma) and 9.1 M urea (Sigma) for 2 hr at 37˚C. Next, we exchanged the solution to ScaleS4 (40% (w/v)) D-sorbitol, 10% (w/v) glycerine (Roth), and 15–25% (v/v) DMSO (Sigma) also at 37˚C. ScalseS4 was renewed after 2 hr and the organoids were maintained at 37˚C until analysis. This is critical, as the clearing effect decreases visibly after 1 hr at room temperature.

## X-Clarity clearing

We performed X-Clarity tissue clearing using an X-CLARITY Polymerization System (Logos biosystems) per manufacturer's instructions. First, organoids were fixed with 4% PFA overnight at 4˚C. The next day, the organoids were embedded in a hydrogel monomer solution and incubated 12–24 hr, followed by a 3-hr polymerization step at −90 bar and 37˚C in the X-CLARITY Polymerization System. The embedded organoids were transferred to a tissue container and cleared in the electrophoretic tissue chamber for 2 hr at 1.2 A and 37˚C.

## ClearT clearing

For ClearT tissue clearing we followed a protocol by *Kuwajima et al., 2013*. In short, fixed organoids were incubated for 30 min at room temperature in increasing formamide (AppliChem) concentrations: 20%, 40%, 80%, and 95% (AppliChem). Samples were directly used for further applications or stored at 4˚C.

## CUBIC clearing

CUBIC tissue clearing was modified after *Susaki et al., 2015*. Briefly, we incubated the organoids for 6 hr at room temperature in CUBIC-1 ½ (one part water and one part CUBIC-1: 25% (w/w) quadrol (Sigma), 25% (w/w) urea, and 15% (w/w) triton X-100 in water). Afterwards, samples were

incubated for 24 hr in CUBIC-1 at 4°C and 24 hr in CUBIC-2 (25% (w/w) urea, 50% (w/w) sucrose, and 10% (w/w) triethanolamine (Sigma)).

## DAPI detection depth for evaluation of clearing efficiency

We stained the organoids with 0.5 μg/mL DAPI (Sigma) in PBS for 24 hr and subsequently subjected them to the different tissue clearing protocols. Afterwards, a LSM 700 scanning confocal microscope (Zeiss) was used to acquire z-stacks of the stained and cleared organoids. Three XZ and three YZ cross-sections per aggregate (n = 10 aggregates per clearing method) were used to quantify the maximum depth at which the DAPI signal could still be detected at a given brightness threshold. The depth of each cross section was measured manually at n = 5 different positions for each slice and n = 10 organoids per clearing protocol using ImageJ/Fiji. The data was exported to Microsoft Excel for further processing and GraphPad Prism v7.0 for plotting and statistical analysis.

## Light transmission analysis for evaluation of clearing efficiency

Following clearing, we took brightfield images of the organoids (n = 10 per protocol, n = 6 for X-Clarity) using an upright stereomicroscope (Leica MZ10 F, camera: Leica DFC425 C) under standardized brightfield transmission conditions. The mean brightness of the organoid area, measured by ImageJ/Fiji, served as measure for the amount of light transmitted through the aggregate. The data was exported to Microsoft Excel for further processing and GraphPad Prism v7.0 for plotting and statistical analysis.

## Statistical analysis for clearing quantification

For statistical analysis of the clearing protocol comparisons, we used GraphPad Prism v7.0 and performed unpaired, two-tailed t-tests with $\alpha = 0.05$ as normal distribution and equal variances can be assumed for the analyzed data.

## Electron microscopy

For electron microscopy analysis, we used n = 3 AMOs from one batch at day 32. Samples were initially fixed for 3 hr with 2% glutaraldehyde, 2% paraformaldehyde in 0.2 M cacodylate buffer, pH 7.2. Afterwards, the specimen was dissected into smaller pieces and post-fixed with 1% osmium tetroxide containing 1.5% potassium hexacyanoferrate. Samples were dehydrated stepwise, including an over-night uranyl-block staining step in 70% ethanol. The specimen was orientated and flat embedded in epon. In total, three samples sectioned under different angles were analyzed at the electron microscope to visualize different aspects of the AMOs.

## Generation of iPSC-based cerebral organoids

For iPSC-derived cerebral organoid generation, we followed the protocol by *Lancaster et al., 2013*. with minor modifications. Briefly, we dissociated iPSCs to single cells by accutase treatment and plated 9000 cells per well in a conical 96-well plate in low FGF stem cell medium (DMEM-F12 with knockout serum replacement (KOSR, Thermo Fisher) 1:5, fetal bovine serum (Biochrom) 1:33.3, 1% penicillin/streptomycin/glutamine, 1% non-essential amino acids (NEAA, Sigma-Aldrich), 2-mercaptoethanol (Thermo Fisher) 1:143, 4 ng/μL FGF2, 50 μm ROCK inhibitor Y-27632, and 0.4% PVA on seeding day only to facilitate aggregation). We exchanged the medium every other day, FGF2 and Y-27632 were withdrawn on day 6. Neural induction was started on day 8 (neural induction medium: DMEM-F12 with KOSR 1:5, 1%penicillin/streptomycin/glutamine, 1% non-essential amino acids, N2 supplement 1:100, and Heparin (Sigma-Aldrich) 1 μg/mL) and continued for 6 days with media changes every other day. On day 13, we embedded the aggregates into 30 μL matrigel droplets and transferred them to 6 cm$^2$ suspension tissue culture dishes (Sarstedt) in cerebral organoid differentiation medium (DMEM-F12 and Neurobasal 1:1 with 1% penicillin/streptomycin/glutamine, 1% NEAA, N2 upplement 1:200, B27 supplement without vitamin A 1:100, Insulin (Sigma-Aldrich) 1:4000, and 2-mercaptoethanol 1:285714). We placed the culture dishes on a shaker at 37°C and 5% $CO_2$ and fed the organoids every other day. On day 20 the B27 supplement was replaced by B27 with Vitamin A (Thermo Fisher) and organoids were cultured until day 30 or 45.

## Initial cleaved caspase 3-based detection of G418 toxicity (*Figure 8—figure supplement 1a–f*)

At day 50, we treated AMOs with increasing concentrations (0, 5, 50, 100, 250, 500, 1000 µg/mL) of G418 (Sigma-Aldrich) added directly to the culture medium. After 2 days, we renewed the medium (including identical toxin concentrations) and fixed the aggregates after a total of 4 days of treatment. Fixation, whole mount immunostaining for cCasp3 and Sox2 as well as BABB-based clearing was performed as outlined above. Image analysis followed the steps as outlined in the high-content analysis section with slight modifications to accommodate the individual brightness, morphology, and background characteristics of the cCasp3 staining. Briefly, after identifying AMOs and Sox2+ cells as described previously, the cCasp3 channel was background corrected by running a sliding parabola algorithm with a curvature setting of 10 across each confocal slice of the AMO. We identified apoptotic cells via the 'find nuclei' function in the 647 nm channel, algorithm 'M' and further selected them to be cCasp3+ if they were larger than 11 µm$^2$, smaller than 100 µm$^2$, and brighter than 2700 abu. We considered cells to be Sox2/cCasp3 double-positive if they fulfilled the criteria for both filters at the same time. The results were outputted to Microsoft Excel, reformatted and then transferred to GraphPad Prism v8.0.2 for plotting, data analysis, and curve fitting.

## Additional G418 toxicity testing with a broader range of concentrations (*Figure 8—figure supplement 1g–i*)

On day 30 of differentiation, AMOs received Tox medium (TM) overnight to remove antioxidants present in the B27 media supplement. TM consisted of DMEM-F12 supplemented with 1% N2 and 1% penicillin/streptomycin/glutamine. On day 31, we added different concentrations of G418 (ranging from 0 to 10,000 µg/mL) dissolved in TM and renewed the medium (including the different compound concentrations) once two days later. After a total treatment time of 96 hr, the samples were analyzed by their size, the CellTiter-Glo 3D Cell Viability Assay, and cCasp3 staining as described before. In the case of cCasp3 staining, the image analysis parameters were used with minor modifications to account for the individual characteristics of the staining, for example signal intensity, aggregate size, background. The calculation of the organoid size/area plotted in *Figure 8—figure supplement 1h* was performed on the same fluorescence images as the cCasp3 quantification in subfigure g) according to the methods outlined in the description of the high content analysis. Data was transferred to Microsoft Excel for further analysis and GraphPad Prism v8.4.2 for plotting and curve fitting.

## Acknowledgements

We thank Britta Trappmann and Martin Weiss for their help with the spinning disc confocal imaging for the calcium activity experiments, as well as Laura Gonzales Cano, Yotam Menuchin-Lasowski, Michele Boiani, and Eva Kutejova for helpful discussion of the manuscript.

## Additional information

### Competing interests

Henrik Renner, Martha Grabos, Mandy Otto, Hans R Schöler, Jan M Bruder: The work presented in this study is the subject of the patent application EP 18 19 2698.0-1120 to the European Patent Office, where HR, MG, MO, HRS, and JMB are inventors. The other authors declare that no competing interests exist.

### Funding

| Funder | Grant reference number | Author |
| --- | --- | --- |
| H2020 European Research Council | grant agreement No [669168] | Henrik Renner<br>Martha Grabos<br>Mandy Otto<br>Dagmar Zeuschner<br>Hans R Schöler<br>Jan M Bruder |

The funders had no role in study design, data collection and interpretation, or the decision to submit the work for publication.

#### Author contributions
Henrik Renner, Conceptualization, Data curation, Formal analysis, Validation, Investigation, Visualization, Methodology, Writing - original draft, Writing - review and editing, Designed and carried out experiments, interpreted results, and wrote the manuscript; Martha Grabos, Investigation, Methodology, Performed iPSC-derived cerebral organoid culture and contributed to quantitative real time PCR analysis and RNA sequencing; Katharina J Becker, Formal analysis, Investigation, Visualization, Methodology, Performed organoid viability measurements and contributed to AMO culture, quantitative real time PCR, Dopamine ELISA, and toxicity evaluation; Theresa E Kagermeier, Investigation, Methodology, Performed automated cortical hiPSC organoid culture and contributed to their analysis; Jie Wu, Data curation, Formal analysis, Visualization, Performed the RNA sequencing analysis; Mandy Otto, Investigation, Methodology, Performed the comparison of clearing protocols and contributed to the quantification of the optical analysis workflow; Stefan Peischard, Paul Disse, Investigation, Methodology, Performed the multielectrode array measurements and analyzed the data; Dagmar Zeuschner, Investigation, Visualization, Performed the electron microscopy analysis; Yaroslav TsyTsyura, Formal analysis, Investigation, Visualization, Methodology, Performed patch-clamping and analyzed the data; Jürgen Klingauf, Investigation, Performed patch-clamping and analyzed the data; Sebastian A Leidel, Formal analysis, Contributed to the RNA sequencing analysis and writing of the manuscript; Guiscard Seebohm, Formal analysis, Visualization, Performed the multielectrode array measurements and analyzed the data; Hans R Schöler, Conceptualization, Supervision, Funding acquisition, Project administration, Contributed to the conception of the project and interpretation of results; Jan M Bruder, Conceptualization, Data curation, Formal analysis, Supervision, Funding acquisition, Validation, Visualization, Methodology, Writing - original draft, Project administration, Writing - review and editing, Performed the high content imaging analysis, contributed to the conception of the project, data review, interpretation of results, and wrote the manuscript

#### Author ORCIDs
Henrik Renner (iD) https://orcid.org/0000-0002-9621-4569
Stefan Peischard (iD) http://orcid.org/0000-0002-1571-9646
Sebastian A Leidel (iD) http://orcid.org/0000-0002-0523-6325
Jan M Bruder (iD) https://orcid.org/0000-0003-3126-0625

#### Decision letter and Author response
Decision letter https://doi.org/10.7554/eLife.52904.sa1
Author response https://doi.org/10.7554/eLife.52904.sa2

## Additional files

#### Supplementary files
• Supplementary file 1. Source data for the calculation of sample retention efficiency shown in *Figure 1d*.

• Supplementary file 2. Complete List of gene ontology (GO) terms for genes significantly (p<0.001) upregulated (log2 fold change >2) in AMOs compared to published midbrain organoids (*Jo et al., 2016*).

• Supplementary file 3. List of primary antibodies in this study.

• Supplementary file 4. List of quantitative real-time PCR primers in this study.

• Transparent reporting form

#### Data availability
All RNA sequencing data generated by us was deposited to the NCBI GEO database (GSE119060).

The following dataset was generated:

| Author(s) | Year | Dataset title | Dataset URL | Database and Identifier |
|---|---|---|---|---|
| Renner H, Grabos M, Otto M, Wu J, Zeuschner D, Leidel SA, Schöler HR, Bruder JM | 2018 | A fully automated high throughput-workflow for human neural organoids | https://www.ncbi.nlm.nih.gov/geo/query/acc.cgi?acc=GSE119060 | NCBI Gene Expression Omnibus, GSE119060 |

The following previously published datasets were used:

| Author(s) | Year | Dataset title | Dataset URL | Database and Identifier |
|---|---|---|---|---|
| Roost MS, Iperen L, Ariyurek Y, Buermans HP, Arindrarto W, Devalla HD, Passier R, Mummery CL, Carlotti F, Koning EP, Zwet EW, Goeman JJ, Lopes SSMC | 2015 | A human fetal transcriptional atlas | https://www.ncbi.nlm.nih.gov/geo/query/acc.cgi?acc=GSE66302 | NCBI Gene Expression Omnibus, GSE66302 |
| Cukuroglu E, Junghyun Jo | 2015 | Transcriptome profiling of DA neurons, human midbrain-like organoids and prenatal midbrain | https://www.ebi.ac.uk/arrayexpress/experiments/E-MTAB-4868/ | ArrayExpress, E-MTAB-4868 |
| Jaffe AE, Jooheon S, Collado-Torres L, Leek JT, Ran Tao, Chao Li, Yuan Gao, Yankai Jia, Maher BJ, Hyde TM, Kleinman JE, Weinberger DR | 2014 | RNAseq data of 36 samples across human brain development by age group from LIBD | https://www.ncbi.nlm.nih.gov/bioproject/?term=PRJNA245228 | NCBI BioProject, PRJNA245228 |

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
