## [Decision Letter]

**Acceptance summary:**

Adapting organoid cultures, which are highly complex, for high throughput screening applications is an important goal for the field. This Tools and Resources article establishes a protocol for generating and processing midbrain organoids in 96 well plates in an automated fashion, with a high degree of homogeneity between the structures. The resultant organoids express critical structural and transcriptomic features of midbrain, and respond functionally to chemical treatments, opening the door to myriad possibilities for chemical screening in this important tissue lineage.

**Decision letter after peer review:**

Thank you for submitting your article "A fully automated high throughput workflow for 3D-based drug screening in human midbrain organoids" for consideration by *eLife*. Your article has been reviewed by three peer reviewers, and the evaluation has been overseen by Didier Stainier as the Senior Editor. The following individual involved in review of your submission has agreed to reveal their identity: Jessica Young (Reviewer #2).

The reviewers have discussed the reviews with one another and the Reviewing Editor has drafted this decision to help you prepare a revised submission.

Summary:

The manuscript by Renner et al. describes the development of an HTS compatible workflow for the generation of human midbrain organoids starting from partially differentiated iPSC. By using neural precursor cells (smNPC) generated using protocols previously described by some of the authors as starting point, the authors reduce the cellular and organ-like complexity compared to iPSC-derived organoids, but gain in reproducibility in well-to-well sphere production, as measured by size and expression of neuronal markers. The claim is made that the new protocol is superior to other protocols in this field because it produces more homogenous organoids. In general, this is a valuable effort to establish automation friendly protocols to generate reproducible neural organoids for HTS, and this article could be appropriate for a Tools and Resources article. However, all of the reviewers agree that new experiments are required to increase the rigor of the work, support the conclusions of the authors, and clearly demonstrate the advance that the technique produces over current methodologies, in addition to a more detailed evaluation of the new method's limitations.

Essential revisions:

1) In accordance with the general requirements for a Tools and Resources article, as this is not a totally novel technology but rather an improvement of neural organoid HTS, the new method should be properly compared and benchmarked against existing methods used in the field. The reviewers felt that the current manuscript lacks direct comparisons (meaning different protocols compared side by side in the same lab/batch). This is a weakness, especially given the strong emphasis in the paper on how this method is superior to others. Thus, a direct comparison of this protocol to a published neural spheroid/organoid protocol is required. This may include an iPS-to-neuron protocol, or alternatively plain neuronal protocol (e.g. Sirenko et al., 2019). The latter may be most appropriate as the question is how these new organoids are better than existing neuronal spheroids, e.g. for toxicity studies. The comparisons between protocols should ideally include both functional as well as gene expression data measurements taken side by side (not downloaded from the literature datasets, as there can be substantial variability from library to library and lab to lab), and quantified in an HTS-appropriate manner.

2) Because drug screening is the stated purpose of developing such automated organoid-based assay platform, it would have been more informative if the authors would have included data to demonstrate % of mature cell populations and more mechanism-based functional validation with specific pharmacological tools (e.g. dopamine receptor antagonists), in different HTS friendly readouts, in addition to calcium fluorescence signal and MEA, such as dopamine secretion (since these are midbrain organoids with dopaminergic neurons) and cell viability (e.g. CellTiterGlo3D). There is very good characterization of neural progenitor/precursor cells, neurons, and astrocytes in the AMOs, what about oligodendrocytes? Can steps be taken to ensure a representative mix of CNS cell types in the AMOs?

3) Deeper assessment of line to line and batch to batch variability, with more detailed QC analysis and assessment of cells at higher magnification and better quantification, would be valuable in demonstrating the homogeneity and quality of the structures. The authors should comment on % of wells produce organoids that pass QC and then % of automated transfer of AMOs from production plate to clearing/imaging plate, and what % of the wells in an imaging plate do not pass QC because of high fluorescence signal. The authors should better characterize the reproducibility between AMOs from independent cell lines. In Figure 2—figure supplement 2 they describe the protocol on an independent line but simply characterize via morphology. As it is well known that different pluripotent stem cell lines, especially hiPSCs, have inherently different differentiation abilities, quantification of neural and neuronal cell markers should also be provided to show that organoids from independent lines can be directly compared in terms of cellular responses. A qPCR analysis, similar to what is presented in Figure 3 should be done on an AMOs from an independent line.

4) There needs to be a detailed explanation and acknowledgement of the limitations of the new protocol in the text. The use of neural progenitor cells has been reported by others to produce neural spheroids with cortical neurons and astrocytes, spontaneous synchronous calcium waves and action potentials, and with good reproducibility to enable HTS. Therefore, the strategy described here to use NPCs to obtain more reproducible neural aggregates is not entirely novel, although by using the smNPCs and the differentiation protocol outlined, the authors obtain aggregates that include mature neurons (e.g dopamine, others?), neural progenitors, and astrocytes (as measured by IHC and qPCR), with organized cell distribution in different concentric zones and with cellular orientations, including radial arrangements, which therefore classify them as organoids, although not as complex as those previously described starting with iPSC. The authors call this aggregates Automated Midbrain Organoids (AMOs). The authors should acknowledge the sacrifice of complexity that would normally be achieved in iPSC-derived neuronal organoids, resulting from dissociation, and absence of microglia. Starting from smNPCs suggests that the AMOs are nearly entirely derived from neural ectoderm. This precludes some important neural cell types like microglia, which come from mesoderm/hematopoietic lineages. Can the authors comment on whether this is a drawback to their protocol if not all CNS cell types are represented? Or comment on whether a modified automated protocol beginning with pluripotent stem cells and including a step to drive hematopoietic progenitor cells could be included?

5) Spontaneous synchronized calcium waves were detected in the AMOs showing neuronal functional coupling. Functional network activity is also shown in the AMOs using MEA measurements. At least a couple of publications have shown spontaneous synchronized calcium wave and MEA signal from neural spheroids and organoids. (e.g. Cleber et al., 2019, and Sirenko et al., 2019), so we don't believe this is the first-time oscillatory waves have been observed in neural aggregates, as the authors state. In general, the authors should retract priority claims ('first time this has been done').

[Editors' note: further revisions were suggested prior to acceptance, as described below.]

Thank you for resubmitting your work entitled "A fully automated high throughput workflow for 3D-based drug screening in human midbrain organoids" for further consideration by *eLife*. Your revised article has been evaluated by Didier Stainier (Senior Editor) and a Reviewing Editor.

The manuscript has been improved but there are some remaining issues that need to be addressed before acceptance, as outlined below:

The consensus of the reviewers is that the authors have addressed most of the issues raised by the reviewers by the including relevant new data, as was requested. The authors have greatly strengthened this paper. They provide adequate discussion of the limitations (lack of other neural cell types, i.e. microglia and oligodendrocytes) and provide a more detailed analysis of midbrain-relevant cell types and measurements of toxicity. We also appreciate the more detailed description of the MEA recordings as this will be useful for other groups attempting similar experiments. For a Tools and Resources article, this is a protocol that will be useful in screening applications.

The reviewers have also identified a number of areas that require revision. In particular it is critical that the authors tone down some of the language described in their estimation of the protocol, relative to other protocols and approaches in the field. The reviewers are not wholly convinced that the Pasca lab organoid differentiation protocol was successfully adopted and scaled up, and the comparisons may be affected by this. There are also numerous minor errors within the manuscript that require attention.

A short list of important revisions is described below. Addressing these will greatly improve the manuscript and make it more suitable for consideration as a published manuscript.

1) Figure 4A-D, F-G need to include units in the y-axis.

2) Figure 8E-J, as displayed, it is not possible to distinguish any effects by the treatments in any of the panels; perhaps the Fourier transform should be displayed to better illustrate the effects of the treatments and the currently shown traces should be moved to a supplemental figure.

3) Figure 8K, it should indicate what MEA measurement was used as "summed absolute signal", and be good to include errors bars and indication of statistical significance between the conditions.

4) Figure 2—figure supplement 2 needs to include concentration units in the dose response plots: log(concentration) in ug/ml?

5) Nice to have but not required. In Figure 2—figure supplement 2: For HTS for toxic effects, it is important to establish readouts that are robust, fast and amenable to screening; CellTiterGlo is one of them and it would also be interesting to see whether G418 induced a reduction in spheroid size, as measure by brightfield microscopy?

6) The analysis and comparison between other, commonly used brain organoid protocols is clearer than before. However, the Sloan et al. protocol part raises some major questions. How efficient was that protocol? It is said to have been performed with modifications. Did the authors attempt the original, unmodified protocol as a positive control? To my limited understanding some of the differences include;

a) Growing cells feeder free to begin with instead of on MEFs (is this the case?).

b) Harvesting with accutase instead of dispase which ends up dissociating the iPS cells instead of forming embryoid body like structures (this is a big difference!).

c) Not using the neural induction media described in that paper.

As the Sloan et al. protocol's performance is presented as evidence of the authors' claim that their protocol is an improvement over other protocols, this really needs to be carefully stated. The best course of action would be to clearly delineate the modifications made, compared to the original protocol, for instance in a supplemental figure. The authors should also tone down their conclusions and make the point that the modifications may have affected the other protocol. One difference is at the detachment step for the cultures – in the Sloan et al. protocol, this is at the iPS cell stage, whereas in the current manuscript the detachment/dissociation step is at the neural progenitor stage. This is likely to benefit the uniformity of the structures/cultures, but may also result in simpler structures than doing the whole protocol from start to finish with the same plate of iPS cells. Making this point in the paper and acknowledging the limitation of starting with a homogenized neural progenitor culture would be worthwhile. In general, this will not detract from the authors' data but will make their conclusions more accurate.

7) The clarification of batches and comparison with an independent cell line is helpful in the conclusion that this is a highly reproducible protocol. But we have concerns about reporting the efficiencies as % without any errors. The authors do not explain where these numbers come from and what they mean, and how they were calculated. If there is a set of reproducible data to back these up, it should be provided, otherwise the numbers should be clearly described as one representative sample experiment rather than a general assessment.

8) The figures and panels should be referenced in order within the text (e.g. the first reference in the Results section is currently to Figure 7 which is an odd way to start). Please ensure that this is adhered to throughout the paper, as there is a lot of data here and it can be confusing otherwise.

9) "The addition of ectopic matrices, including Matrigel, is known to introduce a large amount of variability." The authors suggest that Matrigel introduces variability, but don't show any data to support this. Nor has Matrigel been completely eliminated, as the iPS cells must start out on matrix. As Matrigel is a commonly used ingredient, claims regarding its effects need to be carefully controlled for, rather than relying on citations from the literature. The authors should tone down claims of Matrigel-induced variability for this paper, and simply say that they drew a comparison to another protocol that does not use Matrigel, similar to the one they have developed.

10) In the legend for Figure 4K:

"k) Fast Fourier Transformation (FFT) based on the data shown in k." I believe it is based on the data shown in j, not k.

---

## [Author Response]

Essential revisions:1) In accordance with the general requirements for a Tools and Resources article, as this is not a totally novel technology but rather an improvement of neural organoid HTS, the new method should be properly compared and benchmarked against existing methods used in the field. The reviewers felt that the current manuscript lacks direct comparisons (meaning different protocols compared side by side in the same lab/batch). This is a weakness, especially given the strong emphasis in the paper on how this method is superior to others. Thus, a direct comparison of this protocol to a published neural spheroid/organoid protocol is required. This may include an iPS-to-neuron protocol, or alternatively plain neuronal protocol (e.g. Sirenko et al., 2019). The latter may be most appropriate as the question is how these new organoids are better than existing neuronal spheroids, e.g. for toxicity studies. The comparisons between protocols should ideally include both functional as well as gene expression data measurements taken side by side (not downloaded from the literature datasets, as there can be substantial variability from library to library and lab to lab), and quantified in an HTS-appropriate manner.

We appreciate the reviewers’ demand for additional rigorous comparison and have conducted extensive additional experiments with over 10,000 organoids total. In order to have a stringent and fair comparison, we stipulated a number of requirements:

1) We wanted to compare protocols using the same starting cell lines, since differentiation outcomes may vary widely between different cell lines, as the reviewers have mentioned themselves.

2) To accommodate for the concerns raised by the reviewers regarding lab to lab variability, we wanted to automate the comparison protocol on our platform in house, so that differences in variability are inherent to the protocol and do not stem from other factors including manual handling, thawing, passaging, etc.

3) Ideally, the comparison protocol does not use matrigel, as it is known to induce high variability (Hughes et al., 2010, Proteomics; Jabaji et al., 2014, PLoS One; Tong et al., 2018). We felt that it would be more straightforward to compare other ECM-free protocols to ours for the most conservative comparison possible.

These requirements exclude the use of cortical spheroids as used by Sirenko et al., as they used commercially available spheroids which can only be bought in an already aggregated state/as a ready-to-use product. The company selling these spheroids does not make their protocols publicly available, preventing us from replicating their work in house and using the same starting cell lines for a fair comparison. On a technical level, the spheroids from Sirenko et al. do not possess any discernable self-organization, but are merely 2D-differentiated, re-aggregated neural cultures. Our AMOs are considerably more complex than this, showing distinct self-organized architectures which bring them up to par with organoid rather than spheroid protocols.

Thus, we chose to compare our AMOs to another widely used and accepted organoid protocol from the Pasca lab (Sloan et al., 2018), which has several advantages:

1) We can use the same starting cell line that gave rise to our AMOs.

2) Pasca organoids do not require the addition of matrigel to properly mature.

3) We managed to fully automate the entire generation and analysis of the Pasca organoids in house, using the exact same platform as our organoids.

According to the reviewers’ requests, we then compared this established protocol to our protocol side-by-side in terms of:

- gene expression (Figure 3—figure supplement 1)

- functionality/variability:

> overall morphology and size (Figure 7A,B,E and G)

> viability of single organoids (“CellTiterGlo3D”, Figure 7F and H)

> cellular architecture/structure and quantitative marker expression by high content imaging analysis (Figure 7C,D,I and J, and Figure 7—figure supplement 1)

> electrical activity (Figure 8H,I and J)

- suitability for toxicity testing (Figure 8—figure supplement 1).

We can quantitatively demonstrate that the widely accepted hPSC-based organoid protocol by Sloan et al. has a significantly larger degree of variability than our AMO protocol even when fully standardized and automated.

We would like to point out that the adaptation of the protocol by Sloan et al. into a freely scalable workflow is a significant achievement in itself, and has not been reported in the literature to our knowledge.

2) Because drug screening is the stated purpose of developing such automated organoid-based assay platform, it would have been more informative if the authors would have included data to demonstrate % of mature cell populations and more mechanism-based functional validation with specific pharmacological tools (e.g. dopamine receptor antagonists), in different HTS friendly readouts, in addition to calcium fluorescence signal and MEA, such as dopamine secretion (since these are midbrain organoids with dopaminergic neurons) and cell viability (e.g. CellTiterGlo3D). There is very good characterization of neural progenitor/precursor cells, neurons, and astrocytes in the AMOs, what about oligodendrocytes? Can steps be taken to ensure a representative mix of CNS cell types in the AMOs?

We agree with the reviewers that our manuscript benefits from additional functional characterization of our AMOs. We have carried out a number of additional analyses to bolster this aspect of our manuscript.

- We have quantified the number of mature cells in our organoids for two separate cell lines and 3 independently thawed and cultured batches each (Figure 8D).

- We have demonstrated functional characteristics of midbrain identity using pharmacological agonists and antagonists for different neuronal subpopulations (Figure 8E-K).

- We demonstrate dopamine release into the supernatant via ELISA with levels that match or exceed those found in cerebrospinal fluid in vivo (Figure 8C).

- We have quantified cell viability via CellTiterGlo3D, demonstrating superior homogeneity when compared to hIPSC-derived organoids (Figure 7F).

- We have separately quantified the survival of dopaminergic neurons and other neural populations after specific pharmacological ablation of dopaminergic cells. This clearly underlines the relevance and suitability of our model system for setting up screens for Parkinson’s disease, for example (see Figure 8A and B).

Oligodendrocytes arise very late in neural development, and other protocols demonstrate presence of oligodendrocytes after 100 days of differentiation or more (for example, see Marton et al., 2019). These long time scales are not very attractive for the design of screening strategies. While our smNPCs are capable of efficiently generating oligodendrocytes, as has been published by colleagues of ours (Reinhardt et al., 2013; Ehrlich at al., 2017), this will not occur within the 30-50 day maturation period suggested for screening from our side. Instead, we favor strategies that ectopically add a known number of oligodendrocytes to our AMOs and thus reach a screenable state much earlier (see our Discussion section) in a system with syngeneic but ectopically added oligodendrocytes.

3) Deeper assessment of line to line and batch to batch variability, with more detailed QC analysis and assessment of cells at higher magnification and better quantification, would be valuable in demonstrating the homogeneity and quality of the structures. The authors should comment on % of wells produce organoids that pass QC and then % of automated transfer of AMOs from production plate to clearing/imaging plate, and what % of the wells in an imaging plate do not pass QC because of high fluorescence signal. The authors should better characterize the reproducibility between AMOs from independent cell lines. In Figure 2—figure supplement 2 they describe the protocol on an independent line but simply characterize via morphology. As it is well known that different pluripotent stem cell lines, especially hiPSCs, have inherently different differentiation abilities, quantification of neural and neuronal cell markers should also be provided to show that organoids from independent lines can be directly compared in terms of cellular responses. A qPCR analysis, similar to what is presented in Figure 3 should be done on an AMOs from an independent line.

We thank the reviewers and have added the requested information:

- QC data with details about the efficiency of the different steps of the workflow can now be found in Figure 1D. Our workflow has undergone many iterations of optimization, and overall, we achieve a sample retention rate of 90% from beginning (organoid seeding) to end (data output from high content imaging). We were quite rigorous in reporting inefficiencies, as the 4.2% of samples lost in the transfer of organoids from culture to imaging plates can be virtually eliminated by simply repeating the automated transfer protocol to catch the small number of remaining “stragglers”.

- We have added quantitative high content analyses for a second, independent cell line and a different hiPSC-derived protocol and compared them side by side in Figure 7. The data confirm our claim of higher homogeneity and reproducibility, even across cell lines.

- We have also quantitatively compared different cell lines and protocols via qPCR and present the data side by side (see Figure 3—figure supplement 1).

We are not sure what the request “assessment of cells at higher magnification” means. We demonstrate single cell resolution in high content settings (Figure 6B and C), and generally provide fluorescence data that provide both overview images of whole organoids (rarely done in the organoid literature due to high variance of structures), and we provide enlargements that facilitate understanding cellular identities and detailed single-cell level morphologies where possible (Figure 2B,D,F,G; Figure 2—figure supplement 1; Figure 2—figure supplement 2B-G; Figure 2—figure supplement 3E-I; Figure 7—figure supplement 1B,F and I; and Figure 8—figure supplement 1A, Video 1). If we can provide further high-resolution images, please specify, and we will provide them.

4) There needs to be a detailed explanation and acknowledgement of the limitations of the new protocol in the text. The use of neural progenitor cells has been reported by others to produce neural spheroids with cortical neurons and astrocytes, spontaneous synchronous calcium waves and action potentials, and with good reproducibility to enable HTS. Therefore, the strategy described here to use NPCs to obtain more reproducible neural aggregates is not entirely novel, although by using the smNPCs and the differentiation protocol outlined, the authors obtain aggregates that include mature neurons (e.g. dopamine, others?), neural progenitors, and astrocytes (as measured by IHC and qPCR), with organized cell distribution in different concentric zones and with cellular orientations, including radial arrangements, which therefore classify them as organoids, although not as complex as those previously described starting with iPSC. The authors call this aggregates Automated Midbrain Organoids (AMOs). The authors should acknowledge the sacrifice of complexity that would normally be achieved in iPSC-derived neuronal organoids, resulting from dissociation, and absence of microglia. Starting from smNPCs suggests that the AMOs are nearly entirely derived from neural ectoderm. This precludes some important neural cell types like microglia, which come from mesoderm/hematopoietic lineages. Can the authors comment on whether this is a drawback to their protocol if not all CNS cell types are represented? Or comment on whether a modified automated protocol beginning with pluripotent stem cells and including a step to drive hematopoietic progenitor cells could be included?

According to the reviewers’ request, we have amended the text in several places including the discussion to acknowledge previously published NPC-based protocols as well as the relative sacrifice of complexity compared to published iPSC organoids. We added a whole section describing the impact of reproducibility for phenotypic screening in the field of 3D culture, and we point out the lack of oligodendrocytes and microglia in our protocol.

While other organoid protocols do, in theory, allow for the formation of microglia “in situ” (Ormel et al., 2018), they are, in practice, not common (Marton et al., 2019; Velasco et al., 2019). Because spontaneous generation of microglia in organoids is so rare, variable, and inefficient, other labs have resorted to ectopically adding them to their organoids (Abud et al., 2017; Lin et al., 2018; Muffat et al., 2019). This approach is absolutely feasible for us but an exploration of ectopic addition of other cell types may be outside the scope of this manuscript. We have added a short discussion on the use of our reproducible platform as a cellular scaffold to systematically add other cell types of interest, including microglia and oligodendrocytes.

We are not sure what is meant by “… resulting from dissociation”. Our protocol does not include any dissociation steps other than generating single cell suspensions for initial organoid aggregation, which are common to all organoid protocols.

5) Spontaneous synchronized calcium waves were detected in the AMOs showing neuronal functional coupling. Functional network activity is also shown in the AMOs using MEA measurements. At least a couple of publications have shown spontaneous synchronized calcium wave and MEA signal from neural spheroids and organoids. (e.g. Cleber et al., 2019, and Sirenko et al., 2019), so we don't believe this is the first-time oscillatory waves have been observed in neural aggregates, as the authors state. In general, the authors should retract priority claims ('first time this has been done').

We thank the reviewers for pointing is out. The claim has been removed.

[Editors' note: further revisions were suggested prior to acceptance, as described below.]

The reviewers have also identified a number of areas that require revision. In particular it is critical that the authors tone down some of the language described in their estimation of the protocol, relative to other protocols and approaches in the field.

We have toned down the language of a number of passages in the text, removing claims of higher estimation of our protocol compared to other protocols.

The reviewers are not wholly convinced that the Pasca lab organoid differentiation protocol was successfully adopted and scaled up, and the comparisons may be affected by this.

Please see detailed discussion below.

There are also numerous minor errors within the manuscript that require attention.A short list of important revisions is described below. Addressing these will greatly improve the manuscript and make it more suitable for consideration as a published manuscript.1) Figure 4A-D, F-G need to include units in the y-axis.

Thank you – we have added the missing labels.

2) Figure 8E-J, as displayed, it is not possible to distinguish any effects by the treatments in any of the panels; perhaps the Fourier transform should be displayed to better illustrate the effects of the treatments and the currently shown traces should be moved to a supplemental figure.

We agree with the reviewers that the changes in MEA traces were hard to discern. We have followed the reviewers’ recommendation and changed the plots to FFTs, which unfortunately also resulted in figures that suffered from a similar lack of clarity. We have thus decided to present the MEA data in a way that visually better separates the different agonist/antagonist treatments and at the same time better represents the temporal relationship between different parts of the experimental procedure. After recording electric field baseline activity for a given organoid, we first added an agonist, and, after recording the signal under the influence of the agonist, an inhibitor to the same sample in the same dish. During this time, we continued recording electric field strength and compared the resulting changes after brief equilibration. By presenting these plots in the same coordinate axis in a temporal sequence from left to right, we hope to make it clear that these measurements stem from one and the same sample and are temporally related, while decluttering the prior overlapping graphs. Our purpose with the traces in Figure 8E-J is to convey a direct impression of the raw electric field activity of the organoids as a representative plot; for quantitative comparison of the MEA traces, we have included all available replicate MEA measurements and plotted absolute changes electric field strength in Figure 8K.

Of note, electric field strength varies dependent on the distance to the recording electrode. The organoids are essentially loosely and randomly attached spheres on the underlying array of fixed 2D electrodes, and their random positioning introduces strong sample-to sample variation of the electric field signal just by the nature of their placement. Hence, to achieve quantitative and comparable measurements, we had to normalize any potential changes of activity to an initial basal recording of the very same sample in the very same location of the MEA dish, taking utmost care not to dislodge the organoid while introducing liquids.

We also measured these organoids after only a brief period of attachment, so that they retain their spherical geometry. This course of experiments is our best approximation to reach this goal. We hope that our new Figure 8E-J now better reflect this.

3) Figure 8K should indicate what MEA measurement was used as "summed absolute signal", and be good to include errors bars and indication of statistical significance between the conditions.

Thank you for pointing this out. The exact nature of quantification for Figure 8K was indeed not clear. The figure shows the sum of the absolute electric field potential oscillations over 15 seconds of time. The rationale behind this is that this would detect both changes in frequency and amplitude of the oscillating signal. This way, even small changes should appear clearly when the signal is summed. The original Figure 8K only graphed the one representative dataset depicted in 8E-J; however, the new version now includes all n’s for all conditions including error bars. We edited the text and figure legend to this end.

We have also amended the corresponding Materials and methods section in order help clear up the details of the MEA experiments and their analysis.

4) Figure 2—figure supplement 2 needs to include concentration units in the dose response plots: log(concentration) in ug/ml?

Thank you for pointing this out. We have added the corresponding labels.

5) Nice to have but not required. In Figure 2—figure supplement 2: For HTS for toxic effects, it is important to establish readouts that are robust, fast and amenable to screening; CellTiterGlo is one of them and it would also be interesting to see whether G418 induced a reduction in spheroid size, as measure by brightfield microscopy?

Size measurements are certainly an interesting, simple, and solid option for readout in HTS settings. We have added organoid size measurements for the dose response experiments to Figure 8—figure supplement 1 as suggested. The clearing procedure renders the organoids almost fully transparent, so that they can no longer be detected in brightfield imaging. However, we routinely measure organoid size via fluorescence microscopy as part of our standard optical analysis pipeline for later signal normalization. Here, we retrieved this data.

In this particular experimental setup, with our biological system and in the time frames and concentrations tested, we did not see any appreciable changes in organoid size.

6) The analysis and comparison between other, commonly used brain organoid protocols is clearer than before. However, the Sloan et al. protocol part raises some major questions. How efficient was that protocol?

The automated generation of iPSC-based organoids was highly efficient; we lost one organoid out of a grand total of 12x 96 well plates over 30 days of culture (also see Results section).

Out of 126 iPSC-derived cortical organoids subjected to high content immunostaining, 120 correctly differentiated towards a neural fate confirmed by MAP2 expression (see Figure 7J). One organoid was lost during processing and 5 did not pass imaging quality control, and MAP2 staining could not be confirmed.

At the RNA level, we confirmed expression of a number of relevant early and late neural and glial markers over time, encompassing a total of 168 organoids arranged in different time points and batches (for details, see Figure 3—figure supplement 1). Of note, these organoids did not undergo any selection before analysis and represent an unbiased sampling.

More detailed analyses of protein expression confirmed expected cortical markers including

FoxG1, Ctip2, TBR1 and TBR2, Satb2 at day 30, 40, and 50 post aggregation (Figure 7—figure supplement 2). This figure shows one representative sample chosen from n=3 immunostained samples, all of which were randomly drawn from 96 well plates. All of the samples showed expression of the markers as shown in Figure 7—figure supplement 2.

We show very early time points for iPSC-based cortical organoids. When we compare our structures (here day 40) with some of the earliest images available from the original publication by Pasca et al., 2015, our morphologies resemble the ones shown there at day 52 (see Figure 7—figure supplement 2B).

It is said to have been performed with modifications. Did the authors attempt the original, unmodified protocol as a positive control? To my limited understanding some of the differences include;a) Growing cells feeder free to begin with instead of on MEFs (is this the case?)b) Harvesting with accutase instead of dispase which ends up dissociating the iPS cells instead of forming embryoid body like structures (this is a big difference!).c) Not using the neural induction media described in that paperAs the Sloan et al., protocol's performance is presented as evidence of the authors' claim that their protocol is an improvement over other protocols, this really needs to be carefully stated. The best course of action would be to clearly delineate the modifications made, compared to the original protocol, for instance in a supplemental figure. The authors should also tone down their conclusions and make the point that the modifications may have affected the other protocol.

All of these are very important points to make, and we thank the reviewers for suggesting a more careful and differentiated approach to our comparisons.

We have followed the reviewers’ recommendations and include a new supplementary figure (Figure 7—figure supplement 1) that details the alterations to the original publications by the Pasca lab. We have amended the text to detail these alterations and their potential impact on the resulting organoids. We have toned down the language pertaining to direct comparisons and clearly state that the modifications may affect the biological outcome (see changes in Results section and Discussion section).

We would have preferred to compare our automated protocol with the original; however, it was not possible to automate the original protocol without amendments. For example, as the reviewers correctly pointed out, we used single cell solutions from feeder-free culture instead of lifted colonies from feeder cell cultures. We acknowledge that starting with a single cell suspension as opposed to colonies alters the biology of the starting conditions and may affect the outcome. However, for automated systems, digested colonies are challenging to handle, as they tend to settle quickly via gravity and it is not possible to deposit a single colony per well with a high degree of confidence. With single cell solutions, we could easily homogenize the starting conditions for each well. We thought it crucial to produce both kinds of organoids in an unbiased, automated, and mechanically standardized fashion to clearly attribute all differences to the biological differences instead of manual handling. We made these changes not to favor our protocol, but to test it in the most rigorous manner available to us and in a format that allows high throughput applications, the central tenet of this manuscript.

One difference is at the detachment step for the cultures – in the Sloan et al. protocol, this is at the iPS cell stage, whereas in the current manuscript the detachment/dissociation step is at the neural progenitor stage. This is likely to benefit the uniformity of the structures/cultures, but may also result in simpler structures than doing the whole protocol from start to finish with the same plate of iPS cells. Making this point in the paper and acknowledging the limitation of starting with a homogenized neural progenitor culture would be worthwhile. In general, this will not detract from the authors' data but will make their conclusions more accurate.

Thank you for the feedback, we have implemented that point in the Discussion section.

7) The clarification of batches and comparison with an independent cell line is helpful in the conclusion that this is a highly reproducible protocol. But we have concerns about reporting the efficiencies as % without any errors. The authors do not explain where these numbers come from and what they mean, and how they were calculated. If there is a set of reproducible data to back these up, it should be provided, otherwise the numbers should be clearly described as one representative sample experiment rather than a general assessment.

We apologize for not including this information earlier. It is now part of both the main text and Figure 1. In the interest of data transparency, we also list the full raw data numbers leading to these conclusions as new Supplementary file 1.

A few comments on the on the number of n’s in this data table:

a) We check culture efficiency (sample retention during seeding, media changes, and fixation) regularly, so we have a lot of data/n’s on this step.

b) The efficiency of the transfer step from v-bottom culture plate to flat-bottom imaging plate is usually not critical for us, as we just run the transfer protocol twice to get 100% transfer efficiency and catch any stragglers. Here, we ran a few plates through the transfer step once for the explicit purpose of generating the “single shot” transfer numbers, hence we have fewer n’s here. We have since run additional transfers and have updated the relevant numbers in the manuscript.

We have the lowest n’s for imaging, as the bulk of our samples are used for a variety of other analyses, e.g. cell titer glo, RNA, calcium, or MEA analyses.

8) The figures and panels should be referenced in order within the text (e.g. the first reference in the Results section is currently to Figure 7 which is an odd way to start). Please ensure that this is adhered to throughout the paper, as there is a lot of data here and it can be confusing otherwise.

We have changed the text as requested.

9) "The addition of ectopic matrices, including Matrigel, is known to introduce a large amount of variability." The authors suggest that Matrigel introduces variability, but don't show any data to support this. Nor has Matrigel been completely eliminated, as the iPS cells must start out on matrix. As Matrigel is a commonly used ingredient, claims regarding its effects need to be carefully controlled for, rather than relying on citations from the literature. The authors should tone down claims of Matrigel-induced variability for this paper, and simply say that they drew a comparison to another protocol that does not use Matrigel, similar to the one they have developed.

We have amended the text as requested.

10) In the legend for Figure 4K:"k) Fast Fourier Transformation (FFT) based on the data shown in k." I believe it is based on the data shown in j, not k.

Thank you for your careful review – we have changed the text accordingly.